# Genetic association of inflammatory marker GlycA with lung function and respiratory diseases

Yanjun Guo [1,2,3] ✉, Quanhong Liu[1,2], Zhilin Zheng[1,2], Mengxia Qing[1,2], Tianci Yao[4], Bin Wang[1,2], Min Zhou[1,2], Dongming Wang[1,2], Qinmei Ke[4], Jixuan Ma[1,2], Zhilei Shan [5] & Weihong Chen[1,2] ✉

Association of circulating glycoprotein acetyls (GlycA), a systemic inflammation biomarker, with lung function and respiratory diseases remain to be investigated. We examined the genetic correlation, shared genetics, and potential causality of GlycA ($N = 115,078$) with lung function and respiratory diseases ($N = 497,000$). GlycA showed significant genetic correlation with FEV1 ($r_g = -0.14$), FVC ($r_g = -0.18$), asthma ($r_g = 0.21$) and COPD ($r_g = 0.31$). We consistently identified ten shared loci (including *chr3p21.31* and *chr8p23.1*) at both SNP and gene level revealing potential shared biological mechanisms involving ubiquitination, immune response, Wnt/β-catenin signaling, cell growth and differentiation in tissues or cells including blood, epithelium, fibroblast, fetal thymus, and fetal intestine. Genetically elevated GlycA was significantly correlated with lung function and asthma susceptibility (354.13 ml decrement of FEV1, 442.28 ml decrement of FVC, and 144% increased risk of asthma per SD increment of GlycA) from MR analyses. Our findings provide insights into biological mechanisms of GlycA in relating to lung function, asthma, and COPD.

Systemic inflammation is associated with impaired lung function in healthy adults as well as in patients with lung disease, however, the mechanisms underlying such associations were poorly understood[1]. Circulating glycoprotein acetyls (GlycA)[2] is a nuclear magnetic resonance (NMR) spectroscopy signal that reflects a composite or overall measure of changes in both, the number and the complexity of N-glycan side chains attached to acute phase reactant proteins (e.g., fibrinogen and C-reactive protein) under states of both acute and chronic inflammation, and therefore have potential in providing insights into disease activity and pathophysiological process[3]. GlycA has been gaining increased interest as a serum biomarker for systemic inflammation and has been found to be useful in improving risk prediction for various diseases, including types of cardiovascular diseases, cancer, and all-cause mortality[4]. However, the associations of GlycA with lung function and respiratory diseases remain to be investigated.

Recently, two studies investigated the association between GlycA and chronic obstructive pulmonary diseases (COPD)[5,6]. Prokić et al. found that higher levels of plasma GlycA were significantly associated with COPD (OR = 1.16 in the discovery and OR = 1.30 in the replication sample). Using the prospective data of Rotterdam Study, they further found that circulating GlycA level was a predictive biomarker for COPD incidence (HR = 1.99) but not for mortality in COPD patients. Their

[1]Department of Occupational and Environmental Health, School of Public Health, Tongji Medical College, Huazhong University of Science and Technology, Wuhan, China. [2]Key Laboratory of Environment and Health, Ministry of Education & Ministry of Environmental Protection, School of Public Health, Tongji Medical College, Huazhong University of Science and Technology, Wuhan, Hubei 430030, China. [3]Department of Epidemiology, Program in Genetic Epidemiology and Statistical Genetics, Harvard T.H. Chan School of Public Health, Boston, MA 02215, USA. [4]Department of Geriatrics, Union Hospital, Tongji Medical College, Huazhong University of Science and Technology, Wuhan, China. [5]Department of Nutrition and Food Hygiene, School of Public Health, Tongji Medical College, Huazhong University of Science and Technology, Wuhan, China. ✉e-mail: yanjunguo@hust.edu.cn; wchen@tjmu.edu.cn

results suggested GlycA was a biomarker of inflammatory pathways in the early stage of COPD, presenting in higher concentrations even before the COPD is clinically diagnosed. Kettunen et al. also reported that elevated GlycA was associated with increased 8-year risk of COPD (HR = 1.58). However, it is unclear the underlying mechanisms and if there is causality in explaining the above observed associations.

By leveraging genetic cross-trait analytic approaches and increasing availability of large-scale genomic data (e.g., UK Biobank data with up to 500,000 participants)[7], studies can not only investigate the association of GlycA with lung function and respiratory diseases, but also evaluate the underlying mechanisms in above association at molecule level, e.g., genetic components. Specifically, cross-trait LD Score regression (LDSC) can estimate the genetic correlation between traits which provide useful etiological insights and help prioritize causal relationships[8], while cross-trait meta-analysis can be used to identify SNP-level shared signal in explaining the observed genetic correlation[9], and two-sample Mendelian Randomization (MR) has been widely used to investigate the existence of causality between two traits[10,11].

Therefore, in the present study, we investigated whole-genome genetic correlation, potential causality, identified shared genetics, and functional enrichment of GlycA with lung function and respiratory diseases using GWAS summary statistics for GlycA, lung function parameters, asthma, and COPD using data from large-scale GWAS consortium. Additionally, since GlycA was reported to be a clinical utility similar or complementary to that of high sensitivity C-reactive protein (hsCRP) but with a higher stability, we therefore extended our genetic analysis to other systematic inflammatory biomarkers or acute phase reactants including white blood cell (WBC), hsCRP, fibrinogen, and albumin to make our findings for GlycA more comparable to these well-established clinical inflammatory biomarkers.

## Results

### Genetic correlation of GlycA and other inflammatory biomarkers with lung function parameters, asthma, and COPD

We first evaluated the shared heritability of GlycA with lung function parameters, asthma, and COPD using cross-trait LDSC (Fig. 1). Generally, GlycA showed moderate genetic correlation with lung function parameters and respiratory diseases. Specifically, GlycA showed significant inverse genetic correlation with two lung function parameters (FEV1: $r_g = -0.14$ [95%CI: $-0.19$, $-0.09$; $P = 1.01 \times 10^{-7}$]; FVC: $r_g = -0.18$ [95%CI: $-0.23$, $-0.13$; $P = 3.54 \times 10^{-14}$]) and significant positive genetic correlation with asthma ($r_g = 0.21$ [95%CI: 0.11, 0.31; $P = 1.93 \times 10^{-5}$]) and COPD ($r_g = 0.31$ [95%CI: 0.16, 0.46; $P = 6.04 \times 10^{-5}$]) after controlling for multiple testing ($P < 0.05/30$), while GlycA was only marginally significant related with FEV1/FVC ($r_g = 0.06$ [95%CI: 0.01, 0.11; $P = 0.02$]) and PEF ($r_g = -0.07$ [95%CI: $-0.12$, $-0.03$; $P = 2.74 \times 10^{-3}$]).

When we extended our analysis to other inflammatory biomarkers or acute phase reactants, hsCRP also showed moderate significant genetic correlation with all the lung function parameters (FEV1: $r_g = -0.22$ [95%CI: $-0.25$, $-0.18$; $P = 1.84 \times 10^{-28}$]; FVC: $r_g = -0.26$ [95%CI: $-0.31$, $-0.22$; $P = 5.52 \times 10^{-34}$]; FEV1/FVC: $r_g = 0.06$ [95%CI: 0.03, 0.09; $P = 6.78 \times 10^{-4}$]; PEF: $r_g = -0.09$ [95%CI: $-0.13$, $-0.06$; $P = 1.88 \times 10^{-7}$]), asthma ($r_g = 0.28$ [95%CI: 0.21, 0.35; $P = 1.48 \times 10^{-15}$]) and COPD ($r_g = 0.40$ [95%CI: 0.31, 0.50; $P = 7.58 \times 10^{-17}$]), WBC showed weak genetic correlation with FEV1 ($r_g = -0.07$ [95%CI: $-0.11$, $-0.04$; $P = 7.97 \times 10^{-5}$]), FVC ($r_g = -0.06$ [95%CI: $-0.10$, $-0.03$; $P = 2.26 \times 10^{-4}$]) and PEF ($r_g = -0.06$ [95%CI: $-0.10$, $-0.02$; $P = 1.17 \times 10^{-3}$]), and albumin showed significant genetic correlation with FEV1 ($r_g = 0.06$ [95%CI: 0.03, 0.10; $P = 8.00 \times 10^{-4}$]), asthma ($r_g = -0.13$ [95%CI: $-0.18$, $-0.08$; $P = 1.60 \times 10^{-6}$]) and COPD ($r_g = -0.11$ [95%CI: $-0.18$, $-0.05$;

### Genetic Correlation of GlycA and inflammatory biomarkers with lung function parameters

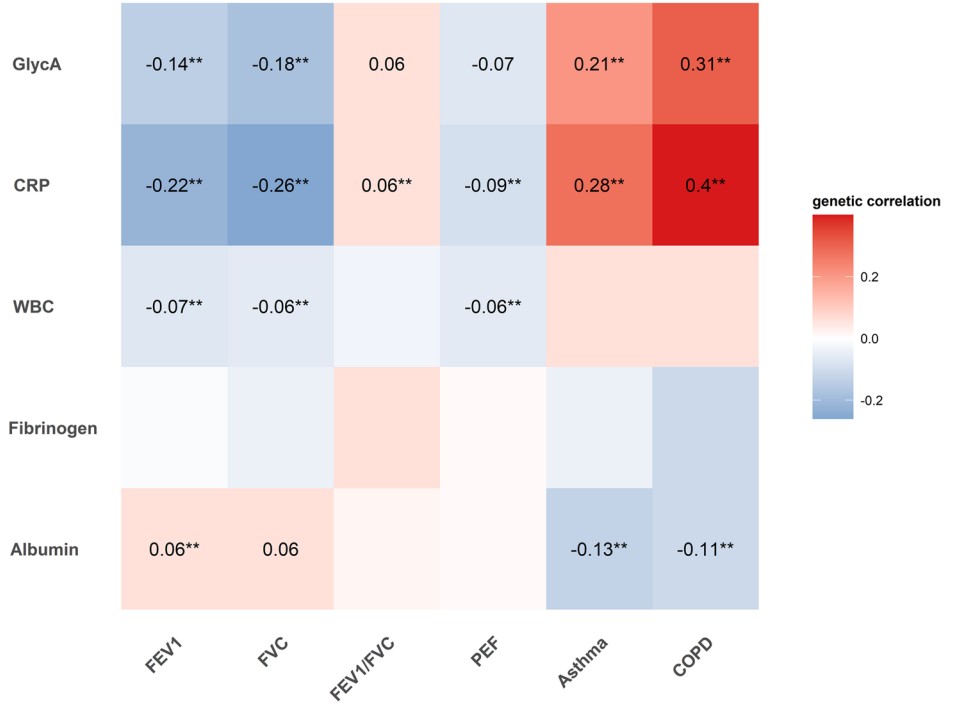

**Fig. 1 | Whole genome genetic correlations of inflammatory biomarkers with lung function parameters (FEV1, FVC, FEV1/FVC ratio, and PEF), asthma, and COPD using linkage disequilibrium score regression (LDSC).** Colors represent the magnitude of genetic correlation of each inflammatory biomarkers with lung function parameters (FEV1, FVC, FEV1/FVC ratio, and PEF), asthma, and COPD using LDSC, red for positive genetic correlation and blue for negative genetic correlation.

Numbers represent the genetic correlation at nominal significance level ($P < 0.05$); ** represent significant genetic correlation after controlling for multiple testing ($P < 0.05/30$). Abbreviations: GlycA glycoprotein acetyls, hsCRP high sensitivity C-reactive protein, WBC white blood cell, FEV1 forced expired volume in 1 s, FVC forced vital capacity, PEF peak expiratory flow, COPD chronic obstructive pulmonary diseases. All $P$ values are two-side.

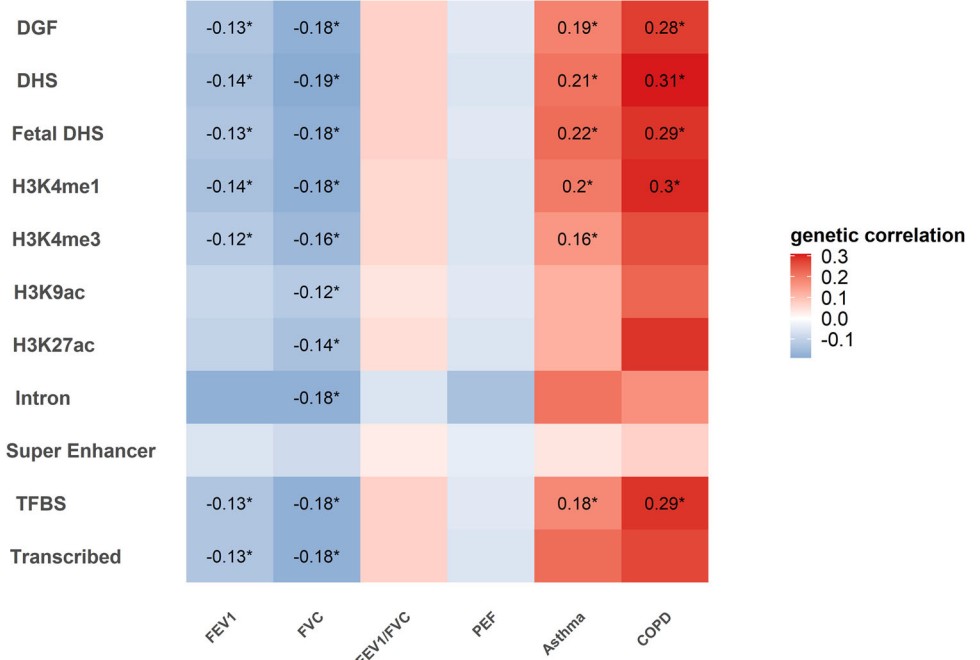

**Partitioned Genetic Correlation of GlycA with Lung Function and Respiratory Diseases**

**Fig. 2 | Partitioned genetic correlations of GlycA with lung function parameters (FEV1, FVC, FEV1/FVC ratio, and PEF), asthma, and COPD according to 11 functional categories using linkage disequilibrium score regression (LDSC).** Colors represent the magnitude of genetic correlation of GlycA with lung function parameters (FEV1, FVC, FEV1/FVC ratio, and PEF), asthma, and COPD at each functional category using LDSC, red for positive genetic correlation and blue for negative genetic correlation. Numbers represent the genetic correlation at nominal significance level ($P < 0.05$); * represent significant genetic correlation after controlling for multiple testing ($P < 0.05/$ [$30 \times 11$]). Abbreviations: GlycA glycoprotein acetyls, FEV1 forced expired volume in 1 s, FVC forced vital capacity, PEF peak expiratory flow, COPD chronic obstructive pulmonary diseases. All $P$ values are two-side.

$P = 3.15 \times 10^{-4}$]). However, fibrinogen showed no significant genetic correlation with any of the lung function parameters and respiratory diseases (asthma and COPD). Sensitivity analysis using GWAS summary statistics from non-overlapping cohorts showed consistent results with the overall genetic correlation. For example, GlycA showed consistent significant inverse genetic correlation with lung function parameters (FEV1: $r_g = -0.17$, $P = 5.84 \times 10^{-5}$); (FVC: $r_g = -0.20$, $P = 3.08 \times 10^{-9}$) (Supplementary Data 2).

**Partitioned genetic correlation of GlycA and other inflammatory biomarkers with lung function parameters, asthma, and COPD**

We next conducted partitioned genetic correlation by 11 functional categories to further explore if the observed significant genetic correlation was attributed to some specific functional region (Fig. 2, Supplementary Figs. 1–4). Generally, results from partitioned genetic correlation analysis were consistent with the overall genetic correlation showing significant genetic correlation of GlycA, hsCRP and albumin with lung function or respiratory diseases at more than one functional category while no significant partitioned genetic correlation for WBC and fibrinogen after controlling for multiple testing ($P < 0.05/$ 330). We observed significant partitioned genetic correlation of GlycA with FEV1, FVC, asthma and COPD at DGF, DHS, fetal DHS, TFBS, and transcribed region (Fig. 2, $r_g$ ranges from −0.12 to 0.31). Significant partitioned genetic correlation of hsCRP with lung function parameters and respiratory diseases were observed across all the 11 functional categories (Supplementary Fig. 2, $r_g$ ranges from −0.08 to 0.41), while significant genetic correlation of albumin was observed to be mainly attributed to DGF, DHS, fetal DHS (Supplementary Fig. 4, $r_g$ ranges from −0.18 to 0.08).

**Cross-trait meta-analysis of GlycA and other inflammatory biomarkers with lung function parameters, asthma, and COPD**

To further explore the existence of shared genetic variants that may underlying the significant genetic correlation observed in our LDSC analysis of GlycA, hsCRP and albumin with lung function or respiratory diseases, we conducted cross-trait meta-analysis of GlycA, hsCRP and albumin with their genetically correlated traits (Supplementary Data 3–5). In total, we identified twelve, fourteen, four and one significant independent shared loci of GlycA with FEV1, FVC, asthma and COPD ($P_{single\ trait} < 1 \times 10^{-5}$ and $P_{meta} < 5 \times 10^{-8}$), respectively. Among these shared loci, eight (*chr3p21.31*, *chr4p15.32*, *chr4q21.3*, *chr6p22.1*, *chr6p21.32*, *chr9q34.2*, *chr16q22.1*, and *chr19q13.32*) were shared across GlycA, FEV1 and FVC. We observed the same shared lead SNP (*rs157595*) of GlycA with FEV1 ($P_{meta} = 1.88 \times 10^{-9}$) and FVC ($P_{meta} = 1.30 \times 10^{-10}$) at locus *chr19q13.32* which is a gene rich region that contains the *TOMM40-APOE-APOC1* genes. Six of above mentioned significant shared loci (*chr1p36.11*, *chr3p21.31*, *chr4p15.32*, *chr6p22.1*, *chr10q21.3*, and *chr19q13.32*) were possible candidate loci for lung function, asthma, or COPD (i.e., these loci were not previously reported to be associated with lung function, asthma, or COPD till we finished the analyses on February, 2023, see details in Supplementary Data 3). Interestingly, five of them (*chr1p36.11*, *chr3p21.31*, *chr4p15.32*, *chr6p22.1*, and *chr10q21.3*) were also identified in the latest lung function GWAS[12]. Although FEV1/FVC ratio and PEF was only marginally significant related with GlycA, we also observed eleven and nine significant shared loci with GlycA, respectively (Supplementary Data 3, $P_{single\ trait} < 1 \times 10^{-3}$ and $P_{meta} < 5 \times 10^{-8}$). Additionally, we observed one locus (*chr8p23.1*) was shared of GlycA with FEV1/FVC ratio (lead SNP: rs6601299, $P_{meta} = 7.52 \times 10^{-23}$), PEF (lead SNP: rs7819276,

$P_{meta} = 3.63 \times 10^{-09}$), asthma (lead SNP: rs113416335, $P_{meta} = 8.52 \times 10^{-09}$) and COPD (lead SNP: rs483916, $P_{meta} = 1.79 \times 10^{-10}$). *CTSB* gene at *chr8p23.1* encodes cathepsin B protein, a member of the C1 family of peptidases.

We then evaluated the shared genetics of hsCRP and albumin with it significantly related lung function parameters and respiratory diseases. In total, we identified 60, 75, 47, 50, 16 and 7 shared loci of hsCRP with FEV1, FVC, FEV1/FVC ratio, PEF, asthma and COPD, respectively (Supplementary Data 4), and we also identified 32, 44, 8 and 2 of albumin with FEV1, FVC, asthma and COPD, respectively (Supplementary Data 5). Notably, the majority of identified shared loci of GlycA with lung function parameters and respiratory diseases were also identified to be shared between hsCRP and at least one lung function parameter or respiratory disease. For example, *chr8p23.1* was also shared of hsCRP with FEV1/FVC ratio, PEF and COPD. We also identified 11 out of the 14 shared loci between GlycA and FVC were shared between hsCRP and FVC which may suggest the similarities of the shared pathways underlying the observed associations of GlycA and hsCRP with lung function and respiratory diseases. In contrast, much less identified shared loci of GlycA with lung function parameters and respiratory diseases were identified to be share between albumin and lung function or respiratory diseases. For example, only one (*chr10q21.3*) out of twelve shared loci between GlycA and FEV1 were identified to be shared between albumin and FEV1.

## Transcriptome-wide association studies

We performed TWAS to identify gene-level genetic overlap of GlycA with its significantly related lung function parameters and respiratory diseases. There were 57, 43, 21, and 28 TWAS significant genes that were identified to be shared of GlycA with FEV1, FVC, asthma and COPD, respectively (Supplementary Data 6–11). Most of these shared TWAS genes were identified from gene expression data in tissues of cardiovascular and nervous system, as well as respiratory system. Restricting this list to shared genes with independent signals after joint/conditional tests, we identified eleven (including *RNU6ATAC3SP* and *HLX* at *chr1q41*, *RNF123* and *UBA7* at *chr3p21.31*, *AFF1* at *chr4q21.3*), ten (including *CFH* at *chr1q31.3*, *RNU6ATAC3SP* and *HLX* at *chr1q41*, *RNF123* at *chr3p21.31*,), three (*IL6R* at *chr1q21.3*, *HPR* at *chr16q22.2*, and *ZNF652* at *chr17q21.32*), and two (*SOX7* at *chr8p23.1* and *TXNL4B* at *chr16q22.2*) significant TWAS genes that were shared of GlycA with FEV1, FVC, asthma, and COPD from tissues including lung, artery, nerve, skin, stomach, esophagus muscularis, and whole blood. Notably, ten of these shared TWAS genes (*chr1q21.3*, *chr1q41*, *chr3p21.31*, *chr4p16.3*, *chr4q21.3*, *chr5q14.1*, *chr5q31.1*, *chr8p23.1*, *chr9q34.2*, and *chr10q21.3*) were also identified from SNP-level cross-trait meta-analysis. *UBA7* at *chr3p21.31* (lead SNP: *rs17650792*) encoded ubiquitin-like modifier-activating enzyme 7, a member of the E1 ubiquitin-activating enzyme family. *AFF1* at *chr4q21.3* (lead SNP: *rs236985*) encoded a member of the AF4/ lymphoid nuclear protein and have been reported to be implicated in human childhood lymphoblastic leukemia.

Additionally, we found *IL6R* at *chr1q21.3* (lead SNP: *rs7529229*) was shared between GlycA and asthma. *IL6R* encodes a subunit of the interleukin 6 (IL6) receptor complex. *SOX7* at *chr8p23.1* encoded a member of the SOX (SRY-related HMG-box) family of transcription factors involved in the regulation of embryonic development and in the determination of the cell fate.

We further extended our analysis to hsCRP and albumin and identified gene-level shared genetics of hsCRP (143, 119, 112, 82, 35, and 30 significant shared TWAS genes with FEV1, FVC, FEV1/FVC ratio, PEF, asthma, and COPD, respectively; Supplementary Data 12–17) and albumin (56, 63, 58, 49, 11, and 7 significant shared TWAS genes with FEV1, FVC, FEV1/FVC ratio, PEF, asthma, and COPD, respectively; Supplementary Data 18–23) with lung function parameters, asthma and COPD. Interestingly, by restricting this list to shared genes with

independent signals after joint/conditional tests, we found that 16 loci and 9 loci that were shared of hsCRP and albumin with lung function parameters, asthma, and COPD were also significant for GlycA (including *chr1q21.3*, *chr1q41*, *chr3p21.31*, *chr4p16.3*, *chr5q14.1*, *chr5q31.1*, *chr8p23.1*, *chr9q34.2*), suggesting potential similar biological pathways in explaining the association of these inflammation biomarkers or acute phase reactant proteins with lung function parameters, asthma, and COPD.

## Functional enrichment analysis

To investigate the biological pathways of the shared genes from cross-trait meta-analysis, we assessed enrichment of shared genes of GlycA with lung function parameters and respiratory diseases in GO biological process. We observed several significant shared pathways of GlycA with lung function parameters and asthma, but not COPD (FDR: $q < 0.05$). Consistent with the gene function of the identified shared loci, GO biological process and KEGG pathway highlighted several common pathways for GlycA sharing with lung function parameters and respiratory diseases, including antigen processing and presentation, inflammatory response, interferon-gamma-mediated and T cell receptor signaling pathway. To further assess the relative enrichment of the shared signals for GlycA with lung function parameters and respiratory diseases in different cell types, we then applied GARFIELD to a generic regulatory annotation denoting open chromatin (DHS) hotspot regions in 424 cell lines and primary cell types from ENCODE3 and Roadmap Epigenomics (Fig. 3). Consistent with the function of the shared signals and pathway analysis, our results from GARFIELD showed predominant enrichment in blood, epithelium, fibroblast, fetal thymus, fetal spleen, and fetal intestine tissues.

Analysis for hsCRP and albumin also showed significant enrichment at pathways including antigen processing, presentation, and inflammatory responses that are shared with lung function parameters, asthma and COPD, while GARFIELD results mainly showed predominant enrichment in blood for hsCRP (Supplementary Fig. 5–10) and albumin (Supplementary Fig. 11–16).

## Mendelian randomization

Finally, we used bi-directional MR instrumental analysis to develop evidence for causality in the relationship of GlycA with lung function parameters and respiratory diseases (Table 1). Genetically instrumented elevated GlycA was associated with increased risk of lung function for FEV1 ($-354.13$ ml, 95% confidence interval [CI]: $-523.90$, $-184.36$; $P = 4.34 \times 10^{-05}$) and FVC (442.28 ml, 95% CI: $-609.85$, $-274.70$; $P = 2.31 \times 10^{-07}$) as well as an increased risk for asthma with odds ratios (OR) of 2.44 (95% CI: 1.47, 4.05; $P = 6.00 \times 10^{-04}$), whereas GlycA was not significantly associated with COPD ($P = 0.99$). Reverse MR showed significant negative instrumental effects of FEV1 ($-62.56$ μmol/L; 95% CI: $-86.59$, $-38.53$; $P = 3.35 \times 10^{-07}$) and FVC ($-69.09$ μmol/L; 95%CI: $-95.95$, $-42.21$; $P = 4.69 \times 10^{-07}$) on GlycA, but asthma was not observed to be significantly related with GlycA ($P = 0.42$).

We then conducted MR instrumental analysis for hsCRP and albumin, and observed that genetically instrumented elevated hsCRP showed significant negative association with FEV1 ($-174.30$ ml; 95%CI: $-212.40$, $-136.21$; $P = 3.03 \times 10^{-19}$) and FVC ($-172.54$ ml; 95%CI: $-211.64$, $-133.45$; $P = 3.03 \times 10^{-19}$) as well as positive association with asthma (OR = 1.36, 95% CI: 1.21, 1.54; $P = 8.84 \times 10^{-07}$) and COPD (OR = 3.18, 95% CI: 2.58, 3.92; $P = 2.14 \times 10^{-27}$), while genetically instrumented elevated albumin showed significant positive association with FEV1 (94.42 ml; 95%CI: 50.57, 138.28; $P = 2.44 \times 10^{-05}$) as well as negative association with asthma (OR = 0.67, 95% CI: 0.59, 0.77; $P = 2.14 \times 10^{-27}$) and COPD (OR = 0.56, 95% CI: 0.45, 0.70; $P = 2.29 \times 10^{-07}$). In addition, reverse MR showed significant negative instrumental effects of FEV1 ($-15.20$ mg/L; 95%CI: $-24.29$, $-1.51$; $P = 1.05 \times 10^{-03}$) and FVC ($-17.48$ mg/L; 95%CI: $-29.08$, $-1.74$; $P = 3.16 \times 10^{-03}$) on hsCRP, whereas we only observed significant negative instrumental effects of per doubling odds of

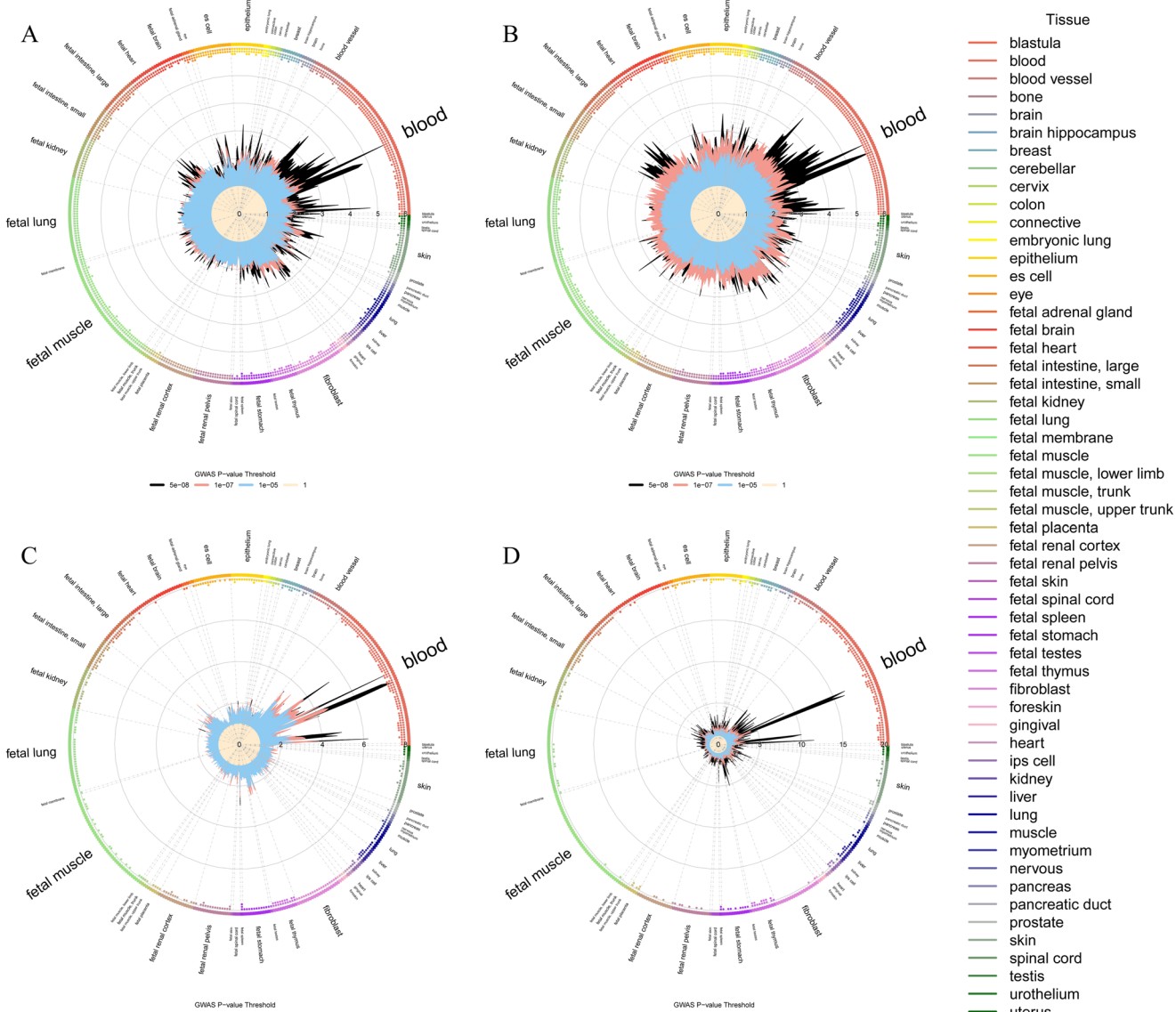

**Fig. 3 | GARFIELD enrichment wheel plots in DNase I–hypersensitive sites (hotspots) for traits (FEV1, FVC, asthma, and COPD) significantly genetically correlated with GlycA. A** GARFIELD enrichment wheel plots for FEV1; **B** GARFIELD enrichment wheel plots for FVC; **C** GARFIELD enrichment wheel plots for asthma; **D** GARFIELD enrichment wheel plots for COPD. Radial lines show OR values at eight GWAS *P* value thresholds (*T*) for all ENCODE and Roadmap Epigenomics DHS cell lines, sorted by tissue on the outer circle. Dots in the inner ring of the outer circle denote significant GARFIELD enrichment (if present) at $T < 10^{-5}$ (outermost) to $T < 10^{-8}$ (innermost) after multiple-testing correction for the number of effective annotations and are colored with respect to the tissue cell type tested (font size of tissue labels reflect the number of cell types from that tissue). GlycA glycoprotein acetyls, FEV1 forced expired volume in 1 s, FVC forced vital capacity, COPD chronic obstructive pulmonary diseases.

asthma (−16.12 mg/L; 95%CI: −22.94, −9.31; $P = 3.53 \times 10^{-06}$) on albumin. Additionally, sensitivity analysis using MR_PRESSO showed consistent results with GSMR, and Steiger directionality test showed that all the causal estimates were oriented in the intended direction (i.e., all $P_{\text{MR-Steiger}} < 0.05$ suggesting that the overall direction of the observed MR associations were correct, Supplementary Data 24). Furthermore, MR results after Steiger filtering showed consistent results with GSMR and suggested that the association of glycA with FEV1 and FVC are more likely to be bidirectional (Supplementary Data 25).

## Discussion

Our genetic analyses were highly consistent and generally support observational associations of positive correlation between GlycA and COPD, but also qualify these associations in important ways: Firstly, besides COPD, we also provided evidence of significant negative association of GlycA with lung function parameters and asthma. Compared

with GlycA, hsCRP showed similar association with lung function, asthma, and COPD; albumin showed weak positive associations, while WBC and fibrinogen was not significantly associated with these traits in overall or partitioned genetic correlation analyses. Secondly, the ten shared loci (*chr1q21.3*, *chr1q41*, *chr3p21.31*, *chr4p16.3*, *chr4q21.3*, *chr5q14.1*, *chr5q31.1*, *chr8p23.1*, *chr9q34.2*, and *chr10q21.3*) identified through both SNP and TWAS analysis revealed potential shared biological mechanisms in GlycA, lung function, and respiratory diseases involving ubiquitination, immune response, Wnt/β-catenin signaling, and cell growth and differentiation in tissues including blood, epithelium, fibroblast, fetal thymus, fetal spleen, and fetal intestine. Finally, we find significant associations of genetically elevated GlycA with lung function and asthma susceptibility from bi-directional MR analyses, which is stronger than those of hsCRP and albumin.

GlycA is an inflammatory marker reflecting both acute and low-grade chronic inflammation, previous studies have demonstrated the

**Table 1 | Bi-directional instrumental estimates of GlycA, hsCRP, and albumin with lung function parameters, asthma and COPD using GSMR**

| Traits | GlycA | | | hsCRP | | | Albumin | | |
|---|---|---|---|---|---|---|---|---|---|
| | Instrumental Estimates | 95%CI | *P* | Instrumental Estimates | 95%CI | *P* | Instrumental Estimates | 95%CI | *P* |
| Forward[a] | | | | | | | | | |
| FEV1 | −354.13 | −523.90, −184.36 | $4.34 \times 10^{-05}$ | −174.30 | −212.40, −136.21 | $3.03 \times 10^{-19}$ | 94.42 | 50.57, 138.28 | $2.44 \times 10^{-05}$ |
| FVC | −442.28 | −609.85, −274.70 | $2.31 \times 10^{-07}$ | −172.54 | −211.64, −133.45 | $5.10 \times 10^{-18}$ | −44.94 | −92.71, 2.83 | 0.07 |
| asthma | 2.44 | 1.47, 4.05 | $6.00 \times 10^{-04}$ | 1.36 | 1.21, 1.54 | $8.84 \times 10^{-07}$ | 0.67 | 0.59, 0.77 | $9.63 \times 10^{-09}$ |
| COPD | 1.00 | 0.42, 2.39 | 0.99 | 3.18 | 2.58, 3.92 | $2.14 \times 10^{-27}$ | 0.56 | 0.45, 0.70 | $2.29 \times 10^{-07}$ |
| Reverse[b] | | | | | | | | | |
| FEV1 | −62.56 | −86.59, −38.53 | $3.35 \times 10^{-07}$ | −15.20 | −24.29, −1.51 | $1.05 \times 10^{-03}$ | −4.03 | −19.81, 11.74 | 0.62 |
| FVC | −69.09 | −95.95, −42.21 | $4.69 \times 10^{-07}$ | −17.48 | −29.08, −1.74 | $3.16 \times 10^{-03}$ | −15.25 | −33.06, 2.55 | 0.09 |
| asthma | −4.72 | −16.32, 6.88 | 0.42 | 0.66 | −2.02, 0.07 | 0.63 | −16.12 | −22.94, −9.31 | $3.53 \times 10^{-06}$ |
| COPD[c] | – | – | – | – | – | – | – | – | – |

GSMR Generalized Summary-data-based Mendelian Randomization, hsCRP high sensitivity C-reactive protein, CI confidence interval, FEV1 forced expired volume in 1 s, FVC forced vital capacity, COPD chronic obstructive pulmonary diseases.

P values are based on two-sided Student's t test and used Bonferroni correction.

[a]The instrumental estimate is corresponding to 1 SD increment of GlycA, hsCRP, or albumin for the forward direction.

[b]The instrumental estimate is corresponding to 1 SD increment of FEV1 or FVC, and per doubling increment of asthma or COPD for the reverse direction

[c]Not enough instruments to conduct reverse GSMR for COPD and inflammatory biomarkers (number of genome-wide significant index SNPs less than 10).

clinical utility of GlycA for predicting incident T2D, CVD, and mortality, independent of hsCRP levels. Bi-directional GSMR suggested that GlycA was causally related with lung function and vice versa. Although in our analyses GlycA showed similar patterns with hsCRP for its genetic associations with lung function, asthma and COPD (i.e., negative association with lung function parameters and positive associations with asthma and COPD), GlycA exhibited stronger association with lung function alteration and asthma (double the instrumental effects of hsCRP on lung function and asthma) from instrumental analyses suggesting potential causality and utility in improving predicting accuracy of lung function and onset of asthma. Since we leveraged germline genetic variation as instrumental variables from large independent studies, our genetic correlation and causal estimates will be less affected by reverse causation and possibly also selection bias as well as environmental and behavioral confounding factors than inference about relationships between GlycA and respiratory diseases from observational epidemiology. GlycA is a composite marker of inflammation, it should be less reactive to acute environmental changes and, therefore, be a more stable measure of chronic inflammation compared to acute-phase markers such as most widely used biomarker-hsCRP[13]. Recently, Crick et al. conducted a study leveraging two large UK population-based cohorts: the Avon Longitudinal Study of Parents and Children (ALSPAC) and the UKB to investigate the stability of GlycA, and their results suggested that GlycA showed greater stability over time than hsCRP and displayed similar or stronger associations with inflammation-related factors (particularly chronic inflammatory states) compared to hsCRP[14]. Additionally, reverse MR showed significant negative effects of FEV1 and FVC on GlycA suggesting that improvement in lung function could also be a potential indicator of systematic inflammation. However, GlycA is a composite measure of inflammation, the clinical utility and specificity as an inflammation biomarker or its potential in translating to future therapies for lung function and respiratory diseases warrant further investigation. Additionally, despite the strong phenotypic association of FEV1/FVC with COPD in observational studies, we observed significant genetic correlation between glycA and COPD but

null for FEV1/FVC. The possible reason for the differences in the magnitude of genetic correlation might be explained by differences in heritability, lack of information on medications, power differences of these two GWASs, and less power for LDSC when genetic correlation is moderate[8,15]. Although sensitivity analysis using independent dataset from SpiroMeta consortium showed consistent results, future studies with larger sample size, repeated measurements and information on respiratory medications are warranted to validate the associations of FEV1/FVC and COPD with glycA.

Additionally, as we observed strong genetic correlation but no significant instrumental effects between GlycA and COPD, it is more likely that the observed associations between GlycA and COPD are explained by shared genetics and related shared etiology rather than causality. Also, the number of instruments for COPD is much smaller than other traits or diseases and the reverse MR analysis might be underpowered.

Meanwhile, using cross-trait meta-analysis and TWAS, we consistently identified ten shared signals (*chr1q21.3*, *chr1q41*, *chr3p21.31*, *chr4p16.3*, *chr4q21.3*, *chr5q14.1*, *chr5q31.1*, *chr8p23.1*, *chr9q34.2*, and *chr10q21.3*) through both SNP and TWAS analysis revealed potential shared biological mechanisms including ubiquitination, immune response, Wnt/β-catenin signaling, and cell growth and differentiation that may be relevant to GlycA, lung function, asthma, and COPD. Specifically, locus *chr3p21.31* was not only identified to be shared across GlycA and multiple lung function parameters, but also suggested to be a candidate locus for lung function and respiratory diseases when compared with previously reported loci. TWAS results further suggested that *UBA7* gene was a potential candidate gene for these traits within this locus, suggesting the signaling pathways regulated by ubiquitination may play roles in lung function. Ubiquitination was reported to play an important role in the pathobiology of lung injury as it regulates the proteins modulating the alveolocapillary barrier and the inflammatory response[16]. Additionally, locus *chr8p23.1* was also identified to be shared across GlycA and multiple lung function parameters/respiratory diseases, but TWAS only validate its association with both GlycA and COPD at *SOX7* suggesting Wnt/β-

catenin signaling pathway through regulating cell proliferation, differentiation, tissue repair and extracellular matrix production in the shared pathology of both GlycA and COPD. The encoding protein of *SOX7* was reported to play a significant role in influencing the factors related to the Wnt/β-catenin signaling pathway, especially the β-catenin[17], while β-catenin signaling by both WNT-dependent and -independent mechanisms plays a key role in airways remodeling of asthmatics[18]. Additionally, cathepsin B protein encoded by *CTSB* gene at *chr8p23.1* was previously reported to be higher among patients with asthma or COPD[19,20]. Furthermore, the identified shared locus at *chr19q13.32* harboring *TOMM40-APOE-APOC1* genes were reported to be expressed by macrophages, alveolar epithelial cells and pulmonary artery smooth muscle cells in the lung, where it may regulate respiratory health and disease[21]. Therefore, our findings supported a potential role of *TOMM40-APOE-APOC1* genes on pulmonary functions attributing to the effect of inflammation through GlycA since elevated GlycA level was found in the apoE ε4+ subgroup[22]. Interestingly, some of these shared signals were also significant for other inflammatory markers, specifically CRP or albumin, which not only suggested that GlycA may capture the glycosylation states of CRP or albumin, but also indicated potential shared biological pathways, such as glycosphingolipid biosynthesis and lipid glycosylation pathways in relating inflammation to lung function and respiratory diseases[23]. However, large prospective studies and randomized trials are warranted to further assess the impact of GlycA in patients with chronic inflammatory conditions including lung function, asthma, and COPD.

In contrast to GlycA and hsCRP, albumin showed positive association with lung function and negative association with asthma and COPD, which is consistent with some epidemiology studies suggesting that people with low serum albumin level had poor lung function, poor clinical outcome and experienced severe bronchiectasis[24]. Our results from genetic analysis consistently suggested that low levels of serum albumin could be a useful indicator of lung function, risk of asthma and COPD. Additionally, the null results of fibrinogen in our study might be a consequence of smaller sample size of fibrinogen GWAS.

This study comprehensively investigates the genetic-based association of GlycA with lung function, asthma, and COPD. The main strengths of our study include large-scale genetic data (sample size up to 497,000), gene-level replication of shared loci from cross-trait meta-analysis by TWAS, and the comparison of multiple well-established acute phase reactants/inflammatory biomarker with the results of GlycA. However, we acknowledged our limitations. Firstly, our study was limited to European ancestry and the shared genetics in other ethnics were uncertain. Therefore, studies in other ethnic groups are encouraged. Secondly, potential sample overlapping may induce bias to genetic estimates. Although sensitivity analysis using GWAS summary-level data from non-overlapping cohorts showed consistent results in our study, future large-scale non-overlapping data are warranted to validate our findings. Thirdly, although we found potential causality and shared mechanisms of GlycA on lung function and onset of asthma, the clinical utility, specificity and potential therapeutic value and of GlycA on lung function and respiratory diseases are warranted for further validation as it is a composite measure reflecting changes in the number and the complexity of N-glycan side chains of acute phase reactant proteins. Fourthly, some of our multiple testing corrections may be overly stringent, because traits included in the present study, such as lung function parameters were highly correlated. Lastly, we are not able to test the added value of GlycA on the basis of hsCRP in predicting the risk of lung function and respiratory diseases in the current study since previous studies suggested that GlycA may capture distinct sources of inflammation and may increase the prediction accuracy in addition to hsCRP for disease risk assessment. Therefore, studies in evaluating the combined effects of GlycA and hsCRP on these traits are encouraged.

In conclusion, the findings further our understanding of the role of GlycA in lung function, asthma, and COPD susceptibility, reveal the prominent genetic-based role and potential shared mechanisms by identifying significant instrumental effects and shared genetic components including *IL6R*, *UBA7*, *CTSB*, *SOX7*, and *TOMM40-APOE-APOC1*, all of which may provide insights into biological mechanisms of lung function, asthma, and COPD.

## Methods

### GWAS summary statistics for GlycA and other inflammatory biomarkers
In the present study, we used the latest large-scale genome-wide summary level data for GlycA ($N = 115,078$) from UK Biobank in October 2022. UK Biobank is a deeply phenotyped cohort of 503,325 participants from the 22 study centers across the United Kingdom (UK) during 2006–2010, and all participants were aged between 40 and 69 at recruitment[7]. GlycA was measured by the Nightingale Health high-throughput NMR platform in approximately one third of randomly selected participants (approximately 121,000 participants). The current analysis used GWAS summary level data for GlycA among all the participants of European ancestry ($N = 115,078$) released in 2020 by UK Biobank (public available at https://gwas.mrcieu.ac.uk/). To further compare the results of GlycA with other systematic inflammatory biomarkers, we extended our analysis to other well-established inflammatory biomarkers or acute phase reactant proteins including WBC, hsCRP, fibrinogen, and albumin. All the summary level data for WBC ($N = 408,112$)[25], hsCRP ($N = 411,229$)[26], fibrinogen ($N = 10,708$)[27], and albumin ($N = 313,032$)[28] were public available at GWAS Catalog (https://www.ebi.ac.uk/gwas/). Details for all the summary statistics can be found in Supplementary Data 1. All the participants were from European ancestry and the genotype data were imputed to the 1000 Genomes reference [29]. This study uses solely GWAS summary statistics, but all participants who contributed to cohorts provided written informed consent and each of the cohort protocols was approved by a local institutional review board.

### GWAS summary statistics for lung function, asthma, and COPD
We used summary-level data for four lung function parameters ($N = 321,047$) which included forced expired volume in 1 s (FEV1), forced vital capacity (FVC), FEV1/FVC, and peak expiratory flow (PEF)[30]. According to the authors providing these summary statistics[30] genome-wide association testing for these traits were performed under an additive genetic model using BOLT-LMM v2.3[31]. GWAS summary-level data for asthma ($N_{cases} = 38,369$, $N_{controls} = 458,631$) and COPD ($N_{cases} = 13,530$, $N_{controls} = 483,470$) were obtained from the National Bioscience Database Center (NBDC) human database (https://humandbs.biosciencedbc.jp/en/) with a total sample size of 497,000[32]. Details for all the summary statistics were summarized in Supplementary Data 1. All the participants were from European ancestry and the genotype data were imputed to the 1000 Genomes ref. [29].

### Genetic correlation of GlycA and other inflammatory biomarkers with lung function parameters, asthma, and COPD using LD score regression
To estimate the shared heritability of GlycA and other inflammatory biomarkers with lung function parameters, asthma, and COPD, we first performed cross-trait linkage disequilibrium (LD) score regression (LDSC, https://github.com/bulik/ldsc/wiki/Heritability-and-Genetic-Correlation, Version 1.0.1) based on GWAS summary statistics to estimate the genetic correlation of GlycA and each inflammatory biomarkers with lung function parameters, asthma, and COPD[33,34]. For LDSC, we used precomputed LD-scores derived from -1.2 million common and well imputed Hapmap3 SNPs in European populations as represented in the Hapmap3 reference panel excluding the HLA

region[33]. The genetic correlation estimates range from −1 to 1. Significance was defined as $P < 0.05/30$ after controlling for multiple testing since we tested the genetic correlation of between five exposures and six outcomes simultaneously. Although it is suggested that cross-trait LDSC is not biased by sample overlap[8], we additionally conducted sensitivity analysis using GWAS summary-level data from non-overlapping cohorts (SpiroMeta consortium) to see if the potential sample overlapping biased the genetic correlation estimation between GlycA and lung function.

### Partitioned genetic correlation of GlycA and other inflammatory biomarkers with lung function parameters, asthma, and COPD

To investigate whether the whole-genome genetic correlation of GlycA and other inflammatory biomarkers with lung function parameters, asthma, and COPD are attributed to specific functional regions, we estimated annotation specific genetic correlations of GlycA and other inflammatory biomarkers with lung function parameters, asthma, and COPD in 11 large annotations using partitioned LDSC. These annotations included DNase I hypersensitivity sites (DHS), fetal DHS, DNaseI digital genomic footprinting (DGF) region, histone marks (H3K4me1, H3K4me3, H3K9ac, and H3K27ac), intron, Super Enhancer, transcription factor binding sites (TFBS), and transcribed region[8,34]. Significance was defined as $P < 0.05/(30*11)$ after controlling for multiple testing since we tested the partitioned genetic correlation of five exposures and six outcomes at 11 functional categories simultaneously.

### Cross-trait meta-analysis of GlycA and other inflammatory biomarkers with lung function parameters, asthma, and COPD

To identify shared genetic variants underlying the observed genetic correlation of GlycA and other inflammatory biomarkers with lung function parameters, asthma, and COPD, we performed SNP-level analysis by conducting pairwise cross-trait meta-analysis using Cross Phenotype Association (CPASSOC)[9] through the statistic SHet that implements a sample-size weighted, fixed effect meta-analysis of the association statistics from combinations of individual traits. Significant shared signals were defined as locus reaching genome-wide significance in joint analysis ($P < 5 \times 10^{-08}$) and reaching suggestive significance for each trait GWAS ($P < 1 \times 10^{-05}$).

### Transcriptome-wide association studies

To identify genes whose expression pattern across tissues implicates etiology or biological mechanisms shared of GlycA with lung function parameters and respiratory diseases, we performed TWAS (http://gusevlab.org/projects/fusion/#installation)[35]. With TWAS, we compared gene-based models of genetic effects on tissue-specific gene expression from GTEx v.7 for GlycA, lung function parameters and respiratory diseases from the GWAS summary statistics to estimate strength of association between concordant gene-based genetic influences on gene expression on GlycA, lung function parameters or respiratory diseases. In total, we performed 48 TWASs for each trait, one tissue-trait pair at a time. We applied Bonferroni correction to identify significant expression-trait associations adjusted for multiple comparisons for all gene-tissue pairs tested for each trait (~200,000 gene-tissue pairs in total, significant expression-trait associations were defined as $P_{Bonferroni} < 0.05$), and then identified genes that had Bonferroni significant associations for both GlycA and lung function parameters /respiratory diseases. We further tested for conditional relationships among the shared genes to identify an independent set of gene-based genetic models using an extension of TWAS that leverages previous methods for joint/conditional tests of SNPs using summary statistics[36].

### Functional enrichment analysis

In order to understand the biological pathways of the significant shared genes from cross-trait meta-analysis ($P_{meta} < 5 \times 10^{-8}$) of GlycA with lung function parameters, asthma, and COPD, we first used the WebGestalt tool[37] to assess overrepresented enrichment of the identified shared gene set in Gene Ontology (GO) biological process pathway. To further explore the functional enrichment of the above identified shared signals, we also conducted GWAS Analysis of Regulatory or Functional Information Enrichment with LD correction (GARFIELD, https://www.ebi.ac.uk/birney-srv/GARFIELD/, Version 2) which leverages GWAS findings with regulatory or functional annotations from 424 cell types and tissues to capture and characterize possible cell-type-specific patterns of enrichment, as provided in the GARFIELD software (primarily from ENCODE and Roadmap epigenomics data) to find features relevant to a phenotype of interest[38].

### Mendelian randomization

To examine the existence of potential causality of GlycA and other inflammatory biomarkers with lung function parameters, asthma, and COPD, we conducted instrumental variable analysis using bi-directional MR implemented in GSMR (https://cnsgenomics.com/software/gcta/#GSMR, Version 1.0.8)[10], which applied a strict criteria to select independent SNP instruments and used heterogeneity in dependent instruments (HEIDI, $P_{HEIDI} < 0.01$) to exclude pleiotropic effect from the analysis. Specifically, genome-wide statistically significant ($P < 5 \times 10^{-8}$) SNPs associated with exposure (i.e., inflammatory biomarkers, lung function parameters or respiratory diseases) with excluding SNPs that were in LD using a criterion of r2 ≥ 0.1 were used as instruments. The remaining LD between the instruments could be resolved using the GSMR method. HEIDI is a method to detect pleiotropic SNPs at which the estimates of $b_{xy}$ (the estimate of the effect of x on y) are significantly different from expected under a causal model, and remove them from the GSMR analysis, with accounting for LD among SNPs using the reference dataset for LD estimation. The individual-level genotypes of the 1000 Genomes Project Phase 3 datasets were utilized as the reference sample to estimate LD between the SNPs. The effects of all instrumental SNP on the exposure and outcome were harmonized to be relative to the same allele. Sensitivity analyses were conducted using Mendelian Randomization Pleiotropy RESidual Sum and Outlier (MR-PRESSO)[39], which identified horizontal pleiotropic outliers in multi-instrument summary-level MR testing and further evaluate the causal effect and direction of GlycA and other inflammatory biomarkers with lung function parameters, asthma, and COPD by minimizing or correcting for horizontal pleiotropy. We also applied Steiger directionality test and Steiger filtering uisng the R package "TwoSampleMR" (https://github.com/MRCIEU/TwoSampleMR, Version 0.5.10) to assure that the causal direction between the hypothesized exposure and outcome was correctly assigned[40]. All the instrumental estimates of GlycA and other inflammatory biomarkers with lung function parameters, asthma, and COPD were interpreted as the estimate changes or odds ratio (OR) per standard deviation (SD) change of GlycA and other inflammatory biomarkers being tested by exponentiating causal estimates for each pair of traits. In addition, since asthma is a binary variable, we interpreted the reverse causal estimates as the average change in GlycA or other inflammatory biomarkers per doubling (2-fold increase) in the odds of asthma, which could be obtained by multiplying the reverse causal estimate by 0.693 ($\log_e 2$)[41]. All the $P$ value are two-sided.

### Reporting summary

Further information on research design is available in the Nature Portfolio Reporting Summary linked to this article.

## Data availability

The summary-level data for GlycA (the UK Biobank [UKB]) used in this study are publicly available at: https://gwas.mrcieu.ac.uk/datasets/met-d-GlycA/; The summary-level data for white blood cell (WBC) used in this study are available at: ftp://ftp.sanger.ac.uk/pub/project/

humgen/summary_statistics/UKBB_blood_cell_traits/ and http://www.mhi-humangenetics.org/en/resources. The summary-level data for high sensitivity C-reactive protein levels used in this study are available at: https://www.ebi.ac.uk/gwas/studies/GCST90079026. The summary-level data for fibrinogen used in this study are available at: https://www.ebi.ac.uk/gwas/studies/GCST90019421. The summary-level data for albumin used in this study are available at: https://www.ebi.ac.uk/gwas/studies/GCST90019493. The summary-level data for lung function parameters used in this study are available at: https://ftp.ebi.ac.uk/pub/databases/gwas/summary_statistics/GCST007001-GCST008000/ with the accession code GCST007429, GCST007430, GCST007431, GCST007432. The summary-level data for asthma and COPD used in this study are available at: https://humandbs.biosciencedbc.jp/en/ with the accession code hum0197, and the GWAS Catalog with the accession code GCST90018795 and GCST90018807, respectively (https://www.ebi.ac.uk/gwas/studies/GCST90018795 and https://www.ebi.ac.uk/gwas/studies/GCST90018807). The individual level data from the UK Biobank (UKB) are available upon application: https://www.ukbiobank.ac.uk/.

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

## Acknowledgements
This research has been conducted using GWAS summary statistics from UK Biobank and FinnGen. We would like to thank the participants and researchers from the UK Biobank and FinnGen who contributed, collected data, or provided the GWAS summary statistics for GlycA, WBC, hsCRP, fibrinogen, albumin, lung function parameters, asthma, and COPD. Y.G. is funded by The National Natural Science Foundation of China (82203995). D.W. is funded by The Science & Technology Fundamental Resources Investigation Program (Grant No.2023FY100604).

## Author contributions
Design the study: Y.G., Z.S., W.C.; Conduct the analysis: Y.G., Q.L., Z.Z., M.Q.; Interpret the results: Y.G., Q.L., Z.Z., M.Q., T.Y., B.W., M.Z., D.W., Q.K., J.M.; Draft the manuscript: Y.G., Z.S., W.C.; Made critical revisions to the manuscript: Y.G., Z.S., W.C.; All authors approved the final version of the manuscript.

## Competing interests
The authors declare no competing interests.
