## [Peer Review File · Nature Communications]

Genetic association of novel inflammatory marker GlycA with lung function and respiratory diseasesREVIEWER COMMENTS

Reviewer #1 (Remarks to the Author):

In this manuscript, Guo et al., leveraged existing large GWAS and conducted genetic analyses to understand the associations between GlycA and lung function and other respiratory diseases. They found significant genetic correlations between GlycA and lung function (FEV1 and FVC), asthma, and COPD. Using cross-trait meta-analysis at SNP and gene level, the authors identified shared loci and potential biological pathways between GlycA and lung function and respiratory diseases. Lastly, Mendelian randomization analysis showed significant instrumental estimates between GlycA and lung function and asthma susceptibility. GlycA is an important inflammatory marker, and understanding its associations with lung function and respiratory diseases is important and clinically relevant. Below are some suggestions:

(1) One major concern for this study is potential sample overlapping. The authors obtained GWAS summary statistics (GlycA, WBC, hsCRP, fibrinogen, albumin, FEV1, FVC, PEF, asthma, and COPD) from UK biobank (or meta-analysis of UKbiobank and FinnGen, Supplementary Table 1), which raise the question of sample overlapping. How it affects the current observation (for example, overfitting) needs to be clarified. The authors could identify non-overlapping cohorts from UKBB for GlycA and lung function (FEV1, FVC) and re-run the analysis. Alternatively, the authors could find additional GWAS from independent cohorts to repeat the study. Furthermore, potential sample overlapping and how it could affect the observations should be discussed in the manuscript.

(2) In this study, the authors also conducted genetic analyses for other inflammatory biomarkers (hsCRP, WBC, albumin, and fibrinogen), which is a strength. However, there needs to be more discussion on the potential relationship between GlycA and other markers. For example, line 339-340, "...16 loci and 9 loci shared of hsCRP and albumin with lung function parameters, asthma and COPD were also significant for GlycA..."—this observation is interesting, and I wonder if any shared pathway involved, or if there is any potential interaction between the biomarkers. Please consider adding more discussion.

(3) What is the rationale for conducting functional enrichment analysis using the shared gene from cross-trait meta-analysis? As ten loci (from the analysis of GlycA) were identified from both cross-trait meta-analysis and TWAS (one of the main findings, Line 397-399), can you conduct functional enrichment analysis using genes within these loci?

(4) Minor suggestions. Some text of IL6R, UBA7, CTSB, SOX7, and TOMM40-APOE-APOC1 in the Results could be moved to the Discussion.

(5) Minor suggestions. Line 434. Not sure if there is enough evidence supporting UBA7 as a "causal gene."

Reviewer #2 (Remarks to the Author):

Global comments

Thank you for the opportunity to read your very detailed paper, which is clearly the product of much work. This study investigates the relationship between an inflammatory marker called GlycA with lung function, and other respiratory traits, using overall and partitioned LD score regression, and two-sample Mendelian randomization.

My key concerns are largely highlighted at least somewhere in their manuscript, which is good, but I do not think they are always given enough weight in the Discussion, and I think the results need more caveating.

My queries about the paper's goals are largely conceptual, related to the utility of GlycA as a biomarker (it seems non-specific?), or as a drug target (because it seems non-specific, and because it could also be a consequence, not a cause of lung function).

I should first caveat my own comments by saying that I am not an inflammation expert. Yet, as the nature of GlycA is that it is a composite measure of inflammation, it strikes me that it could be a difficult candidate to use for risk prediction, given its associations with other chronic diseases such as CVD and T2D, even if it could be proven that the 'forward' relationship dominated (e.g. it preceded respiratory traits – and I am not sure the data show that this is the case, given the strong results from the reverse direction MR). If it is the end-point of multiple inflammatory processes, even if the forward relationship is true, I also think it would be worth giving more discussion on whether it would likely be a feasible drug target – would it not be more fruitful to look upstream for a more specific pathway (which the authors do try to do)?

It doesn't seem particularly surprising to me that cross-trait analysis shows shared biological mechanisms for the association between GlycA and multiple lung function traits, as we already know the pathophysiology of these respiratory traits are correlated, so indeed the pathways involved in inflammation in other disease states will presumably be correlated as well.

The supplement is 278 pages long! I cannot review it in detail.

Specific comments

Abstract

Why look at GlycA beyond the fact it is a systemic inflammatory marker? Can you add a brief sentence to your abstract?

From what I understand (and as I have said, I am not an inflammation biochemistry expert), GlycA may be a marker of multiple inflammatory markers such as TNF-alpha, CRP, fibrinogen, and IL-6. As you state further on, the whole analysis therefore seems very prone to reverse causation, and you need to state in the abstract how you have tried to address this.

Introduction

You have done bidirectional MR which is good, but as MR studies are not immune from reverse causation, I think you should cite a paper which explains this (this is an example, you could choose a different paper): <https://link.springer.com/article/10.1007/s10654-021-00726-8>

Methods

I cannot see a list of variants used in each MR analysis, nor a descriptive of how these were chosen (sorry if I have missed this). Please include this information, along with details of significance thresholds you used for IV selection, and details of any LD clumping r^2 thresholds, allele harmonisation, etc. Please review the STROBE-MR statement for guidance of what to include. <https://www.strobe-mr.org/>

Please note that newer GWAS lung function summary statistics are now available as part of the following publication: <https://www.nature.com/articles/s41588-023-01314-0>. You may need to evaluate the novelty of some of your findings for the loci you identify in light of this (I am not suggesting you re-do analyses).

It is good that you have done bidirectional MR. However, I think it might be important to do further analyses, exploring how your causal estimates change when you remove instruments that are stronger instruments for the outcome than for the exposure (for example, in terms of variance explained by the SNP). Whilst it is not a technique without flaws, Steiger filtering could be one way of exploring this: <https://mr-dictionary.mrcieu.ac.uk/term/steiger/#:~:text=Steiger filtering is a method,that the genetic variant influences.>

It's good to consider the burden of multiple testing – however, you may wish to point out that some of your corrections may be overly stringent, given that the respiratory outcome you study are themselves highly correlated with one another (and your other traits will be correlated too).

Please give more details about the HEIDI method and how it works in terms of 'excluding pleiotropic effects'. My understanding is that it examines whether effect sizes for LD-correlated SNPs in a region are more different than one would expect by chance.

Results

The results section is very verbose with lots of long lists of locus names and numbers of loci. It feels fairly overwhelming to read at times, and I wonder if the section could be shortened, with just key messages highlighted (and the reader directed to figures and tables elsewhere).

Observing shared loci in cross-trait meta-analysis is presumably to be expected – for example, FEV1 and FVC are highly correlated, so I am unsure how interesting it is that the same top SNP comes up for one of the loci mentioned.

Line 273 - novel candidate loci for COPD – please see Shrine et al 2023 which almost quadruples the known number of loci for lung function – are any of the loci mentioned in the paper?

Line 291 – if loci are shared between GlycA and LF and CRP and LF, could it be that CRP is an upstream marker of GlycA?

I find it very surprising that the MR result was so totally null for COPD, given the strong MR results observed for asthma and lung function, and the strong genetic correlation with COPD. These traits are all highly genetically correlated, so to see no effect at all for COPD strikes me as unusual. What explanations

do the authors have for this? In what circumstances would you expect a null MR result (in both directions) in the presence of a strong genetic correlation? It would be important to add this to the text.

Discussion

Can you comment on why you may have found only weak genetic correlation results between GlycA and FEV1/FVC? It seems odd, given the strong correlation with COPD. Moreover, the directions are in the opposite direction to what one might expect, given the association with COPD (higher values of FEV1/FVC associated with inflammation).

Can you comment on why you might have found no correlations with fibrinogen? Do you think this is an artefact of smaller sample size or measurement error, or might there be a biological reason?

As mentioned above, if GlycA is predictive of T2D, CVD AND respiratory disease, it doesn't strike me as a particularly useful predictive marker, as it is non-specific. I can however see that it might be useful in flagging a harmful pro-inflammatory state. If it is indeed true that raised GlycA levels predict and precede disease (and the reverse MR result does make me question this somewhat), what mechanisms do you hypothesise for its elevation? Could it be a marker of smoking, for example?

I am unsure about how GlycA might translate to novel clinical therapies, if it is an end-marker of multiple inflammatory pathways. You acknowledge this in the limitations, but still mention it earlier on in the discussion. Similarly, if ubiquitination is indeed a 'ubiquitous' process in the body, is it likely to be a good drug target? I was interested to see this paper, which suggests it might be possible, although I imagine it would be very fraught with difficulty <https://www.nature.com/articles/cr201631>

REVIEWER COMMENTS

Reviewer #1 (Remarks to the Author):

In this manuscript, Guo et al., leveraged existing large GWAS and conducted genetic analyses to understand the associations between GlycA and lung function and other respiratory diseases. They found significant genetic correlations between GlycA and lung function (FEV1 and FVC), asthma, and COPD. Using cross-trait meta-analysis at SNP and gene level, the authors identified shared loci and potential biological pathways between GlycA and lung function and respiratory diseases. Lastly, Mendelian randomization analysis showed significant instrumental estimates between GlycA and lung function and asthma susceptibility. GlycA is an important inflammatory marker, and understanding its associations with lung function and respiratory diseases is important and clinically relevant.

>>Response: Thank you for your positive remarks on our paper.

Below are some suggestions:

(1) One major concern for this study is potential sample overlapping. The authors obtained GWAS summary statistics (GlycA, WBC, hsCRP, fibrinogen, albumin, FEV1, FVC, PEF, asthma, and COPD) from UK biobank (or meta-analysis of UKbiobank and FinnGen, Supplementary Table 1), which raise the question of sample overlapping. How it affects the current observation (for example, overfitting) needs to be clarified. The authors could identify non-overlapping cohorts from UKBB for GlycA and lung function (FEV, FVC) and re-run the analysis. Alternatively, the authors could find additional GWAS from independent cohorts to repeat the study. Furthermore, potential sample overlapping and how it could affect the observations should be discussed in the manuscript.

>>Response: Thank you for your great suggestions. We now re-ran the analysis for GlycA and lung function using GWAS summary statistics from SpiroMeta consortium with up to 83,080 participants (which is an independent cohort from UK Biobank). Despite the fact that the sample size of SpiroMeta consortium is relatively small compared to that from UK Biobank, the genetic correlations were consistent with the primary results (see the table below for details). We now added sentences in Methods, Results and Discussion to discuss the effects of potential sample overlapping as suggested.

Methods: *“Although it is suggested that cross-trait LD Score regression is not biased by sample overlap (PMID: 26414676), we additionally conducted sensitivity analysis using GWAS summary-level data from non-overlapping cohorts (SpiroMeta consortium) to see if the potential sample overlapping biased the genetic correlation estimation between GlycA and lung function.”* (lines 143 to 147, page 8)

Results: “Sensitivity analysis using GWAS summary statistics from non-overlapping cohorts showed consistent results with the overall genetic correlation. For example, GlycA showed consistent significant inverse genetic correlation with lung function parameters (FEV1: $rg=-0.17$, $P\text{ value}=5.84\times 10^{-5}$]; FVC: $rg=-0.20$, $P\text{ value}=3.08\times 10^{-9}$]) (Supplementary Table 2).” (lines 252 to 156, page 13)

Discussion: “Secondly, potential sample overlapping may induce bias to genetic estimates. Although sensitivity analysis using GWAS summary-level data from non-overlapping cohorts showed consistent results in our study, future large-scale non-overlapping data are warranted to validate our findings.” (lines 513 to 516, page 24)

Trait1	Trait2	rg	se	z	p
GlycA	FVC	-0.20	0.03	-5.93	3.08E-09
	FEV1	-0.17	0.04	-4.02	5.84E-05
	FEV1_to_FVC_RATIO	0.03	0.05	0.55	0.58
	PEF	-0.13	0.08	-1.73	0.08
CRP	FVC	-0.26	0.03	-8.35	6.97E-17
	FEV1	-0.25	0.03	-8.34	7.77E-17
	FEV1_to_FVC_RATIO	-0.02	0.03	-0.46	0.65
	PEF	-0.13	0.05	-2.46	0.01
Fibrinogen	FVC	0.04	0.09	0.38	0.70
	FEV1	0.00	0.10	0.03	0.97
	FEV1_to_FVC_RATIO	0.00	0.14	0.00	1.00
	PEF	-0.28	0.16	-1.72	0.08
Albumin	FVC	0.05	0.03	1.51	0.13
	FEV1	0.07	0.03	2.38	0.02
	FEV1_to_FVC_RATIO	0.03	0.03	0.99	0.32
	PEF	-0.04	0.05	-0.83	0.41
WBC	FVC	-0.06	0.03	-2.08	0.04
	FEV1	-0.09	0.03	-2.99	2.81E-03
	FEV1_to_FVC_RATIO	-0.06	0.04	-1.61	0.11
	PEF	-0.14	0.05	-2.64	0.01

(2) In this study, the authors also conducted genetic analyses for other inflammatory biomarkers (hsCRP, WBC, albumin, and fibrinogen), which is a strength. However, there needs to be more discussion on the potential relationship between GlycA and other markers. For example, line 339-340, “...16 loci and 9 loci shared of hsCRP and albumin with lung function parameters, asthma and COPD were also significant for GlycA...”—this observation is interesting, and I wonder if any shared pathway involved, or if there is any potential interaction between the biomarkers. Please consider adding more discussion.

>>Response: Thank you for your great suggestions. We agree that these overlapping signals may suggest shared pathway or potential interaction. According to your suggestion, we conducted pathway analysis using these shared genes and found that glycosphingolipid biosynthesis and

lipid glycosylation pathways may play a role. We added a sentence to discuss this finding in the Discussion section as suggested. *“Interestingly, some of these shared signals were also significant for other inflammatory markers, specifically CRP or albumin, which not only suggested that GlycA may capture the glycosylation states of CRP or albumin, but also indicated potential shared biological pathways, such as glycosphingolipid biosynthesis and lipid glycosylation pathways in relating inflammation to lung function decline and respiratory diseases.”* (lines 492 to 496, page 24)

(3) What is the rationale for conducting functional enrichment analysis using the shared gene from cross-trait meta-analysis? As ten loci (from the analysis of GlycA) were identified from both cross-trait meta-analysis and TWAS (one of the main findings, Line397-399), can you conduct functional enrichment analysis using genes within these loci?

>>Response: Thank you for your suggestions. The rationale of conducting functional pathway enrichment analysis was to provide insights into the shared biological pathways of the observed significant genetic association of GlycA with lung function and respiratory diseases. In this analysis, we used all the significant shared loci from cross-trait meta-analysis. For example, we identified fourteen significant shared loci between GlycA and FEV1, which resulted in 227 genes within these loci, and identified significant pathways including antigen processing and presentation and innate immune response. We now made it clear that we used significant shared loci from cross-trait meta-analysis to investigate the biological pathways in the Methods (lines 187 to 188, page 10) and Results (lines 370 to 371, page 18) section.

(4) Minor suggestions. Some text of IL6R, UBA7, CTSB, SOX7, and TOMM40-APOE-APOC1 in the Results could be moved to the Discussion.

>>Response: Thank you for your suggestions. We now moved the text of shared genes from Results to Discussion.

(5) Minor suggestions. Line 434. Not sure if there is enough evidence supporting UBA7 as a “causal gene.”

>>Response: Thank you for your suggestions. We revised it as *“UBA7 gene was a potential candidate gene...”*

Reviewer #2 (Remarks to the Author):

Global comments

Thank you for the opportunity to read your very detailed paper, which is clearly the product of much work. This study investigates the relationship between an inflammatory marker

called GlycA with lung function, and other respiratory traits, using overall and partitioned LD score regression, and two-sample Mendelian randomization.

>>Response: Thank you for taking time to review our manuscript and provide valuable comments.

My key concerns are largely highlighted at least somewhere in their manuscript, which is good, but I do not think they are always given enough weight in the Discussion, and I think the results need more caveating.

My queries about the paper's goals are largely conceptual, related to the utility of GlycA as a biomarker (it seems non-specific?), or as a drug target (because it seems non-specific, and because it could also be a consequence, not a cause of lung function).

>>Response: Thank you for your suggestion. We agree that the clinical utility of GlycA is still of uncertainty and warrant further discussion. Low-grade chronic systemic inflammation was reported to be a common mechanistic pathway contributing to reduced lung function and increased respiratory events. However, previous studies mainly focused on high C-reactive protein C-reactive protein (hs-CRP) and interleukin (IL)-6. GlycA is a novel composite biomarker of active systemic inflammation, quantifying both the protein concentrations and glycosylation states of several acute phase proteins. Compared to hs-CRP, GlycA is a more sensitive and a more convenient inflammation marker as it only requires a single measurement. Additionally, GlycA reflects multiple inflammatory pathways, and elevated levels can be seen in both acute and chronic inflammation. Also, hs-CRP arises early in the acute phase response, whereas most proteins that generate GlycA occur later in the acute phase response (PMID: 29078787). Therefore, GlycA appears to be a promising new composite biomarker of active systemic inflammation in patients with inflammatory diseases, and be a complementary marker to other inflammatory markers identified to be associated with lung function decline or respiratory diseases. However, future studies are warranted to further explore its clinical utility. We have added discussion and corrections in Methods, Results, and Discussion accordingly, details were included in the response of the following comments.

I should first caveat my own comments by saying that I am not an inflammation expert. Yet, as the nature of GlycA is that it is a composite measure of inflammation, it strikes me that it could be a difficult candidate to use for risk prediction, given its associations with other chronic diseases such as CVD and T2D, even if it could be proven that the 'forward' relationship dominated (e.g. it preceded respiratory traits – and I am not sure the data show that this is the case, given the strong results from the reverse direction MR). If it is the end-point of multiple inflammatory processes, even if the forward relationship is true, I also think it would be worth giving more discussion on whether it would likely be a feasible drug target – would it not be more fruitful to look upstream for a more specific pathway (which the authors do try to do)?

>>Response: Thank you for your suggestion. We agree that GlycA is a composite measure of inflammation related with multiple chronic diseases and the solely risk prediction value for lung function and respiratory diseases warrant further investigation. However, previous studies suggested that low-grade systemic inflammation markers including hs-CRP, IL-6, blood eosinophils, and blood neutrophils offer independent and additive information in relation to lower FEV1 and FVC in the general population, and indicated that a combination of biomarkers yields more information than the biomarkers assessed individually (PMID: 22427534, 10733371, 30170809). Therefore, since GlycA reflects multiple inflammatory pathways and chronic inflammation status, and may serve as a complementary marker to other inflammatory markers, adding GlycA into a risk prediction model could potentially increase the prediction accuracy.

According to your suggestion, we now rewrote and added discussion regarding whether GlycA/ related pathway would be a feasible drug target or not in the Discussion section. “ *GlycA exhibited stronger association with lung function alteration and asthma (double the instrumental effects of hsCRP on lung function and asthma) from instrumental analyses suggesting potential causality and utility in improving predicting accuracy of lung function decline and onset of asthma. Since we leveraged germline genetic variation as instrumental variables from large independent studies, our genetic correlation and causal estimates will be less affected by reverse causation and possibly also selection bias as well as environmental and behavioral confounding factors than inference about relationships between GlycA and respiratory diseases from observational epidemiology. However, GlycA is a composite measure of inflammation, the clinical utility as an inflammation biomarker or its potential in translating to novel therapies for lung function and respiratory diseases warrant further investigation.*

“ (lines 438 to 450, page 21)

It doesn't seem particularly surprising to me that cross-trait analysis shows shared biological mechanisms for the association between GlycA and multiple lung function traits, as we already know the pathophysiology of these respiratory traits are correlated, so indeed the pathways involved in inflammation in other disease states will presumably be correlated as well.

>>Response: Thank you for your comments. We totally agree that the correlated traits or diseases may share biological mechanisms, pathways or molecular basis. We therefore conducted cross-trait meta-analysis, TWAS and enrichment analysis with the aim of identifying shared SNPs, genes or pathways and providing insights into understanding the mechanisms of comorbid of inflammation, lung function decline and respiratory diseases, which may be helpful for their early diagnosis, treatment, and management, thereby helping reduce the global disease burdens associated with these multi-morbidities.

The supplement is 278 pages long! I cannot review it in detail.

>>Response: Thank you for your comments. We agree that the supplementary file is long, we tried our best to cut down it to 252 pages. However, since we analyzed five inflammatory markers, four lung function parameters and two respiratory diseases, we do have many supporting files, specifically on the shared genes from TWAS.

Specific comments

Abstract

Why look at GlycA beyond the fact it is a systemic inflammatory marker? Can you add a brief sentence to your abstract?

>>Response: Thank you for your suggestions. Beyond the fact it is a systemic inflammatory marker, glycoscience researches show that glycans play an important role in progression of respiratory system disease. According to your suggestion, we added one sentence to the abstract. *“Glycans play an important role in progression of respiratory system disease. Circulating glycoprotein acetyls (GlycA), a novel systemic inflammation biomarker, was useful in improving risk prediction for various diseases, but its associations with lung function and respiratory diseases remain to be investigated.”* (lines 33 to 36, page 3)

From what I understand (and as I have said, I am not an inflammation biochemistry expert), GlycA may be a marker of multiple inflammatory markers such as TNF-alpha, CRP, fibrinogen, and IL-6. As you state further on, the whole analysis therefore seems very prone to reverse causation, and you need to state in the abstract how you have tried to address this.

>>Response: Thank you for your suggestions. We agree that GlycA is a composite measure of inflammation related with multiple chronic diseases and reflects multiple inflammatory pathways, the inferences of relationships between GlycA and respiratory diseases from observational epidemiology studies could therefore be subject of reverse causation. However, we leveraged germline genetic variation as instrumental variables from large independent studies, our genetic correlation and causal estimates will be less affected by reverse causation and possibly also selection bias as well as environmental and behavioral confounding factors. We now added one sentence in the abstract to address this. *“All analyses were less likely to be affected by reverse causation and selection bias as well as environmental and behavioral confounding factors since germline genetic variation from large independent studies were used.”*

Introduction

You have done bidirectional MR which is good, but as MR studies are not immune from reverse causation, I think you should cite a paper which explains this (this is an example, you could choose a different paper): <https://link.springer.com/article/10.1007/s10654-021-00726-8>

>>Response: Thank you for your great suggestions. We agree that we should cite a paper explaining the issues on reverse causation. We have now cited the suggested paper in the Introduction (reference 11)

Methods

I cannot see a list of variants used in each MR analysis, nor a descriptive of how these were chosen (sorry if I have missed this). Please include this information, along with details of significance thresholds you used for IV selection, and details of any LD clumping r^2 thresholds, allele harmonisation, etc. Please review the STROBE-MR statement for guidance of what to include. <https://www.strobe-mr.org/>

>>Response: Thank you for your great suggestions. We agree that it is important to include detailed description of instrumental variables. We now added a description in the methods section. *“Specifically, genome-wide statistically significant ($P < 5 \times 10^{-8}$) SNPs associated with exposure (i.e., inflammatory biomarkers, lung function parameters or respiratory diseases) with excluding SNPs that were in LD using a criterion of $r^2 \geq 0.1$ were used as instruments. The remaining LD between the instruments could be resolved using the GSMR method. HEIDI is a method to detect pleiotropic SNPs at which the estimates of b_{xy} (the estimate of the effect of x on y) are significantly different from expected under a causal model, and remove them from the GSMR analysis, with accounting for LD among SNPs using the reference dataset for LD estimation. The individual-level genotypes of the 1000 Genomes Project Phase 3 datasets were utilized as the reference sample to estimate LD between the SNPs. The effects of all instrumental SNP on the exposure and outcome were harmonized to be relative to the same allele.”* (lines 202 to 212, page 11). Number of instruments were added in Supplementary Table 24.

Please note that newer GWAS lung function summary statistics are now available as part of the following publication: <https://www.nature.com/articles/s41588-023-01314-0>. You may need to evaluate the novelty of some of your findings for the loci you identify in light of this (I am not suggesting you re-do analyses).

>>Response: Thank you for your great suggestions. We compared our novel loci with the newest GWAS and five of them (chr1p36.11, chr3p21.31, chr4p15.32, chr6p22.1, and chr10q21.3) were also identified by the larger GWAS suggesting reliability of the cross-trait meta-analysis in increasing power of identifying novel genetic loci. We have commented on it in the Results section (lines 298 to 300, page 15).

It is good that you have done bidirectional MR. However, I think it might be important to do further analyses, exploring how your causal estimates change when you remove instruments that are stronger instruments for the outcome than for the exposure (for example, in terms of variance explained by the SNP). Whilst it is not a technique without flaws, Steiger filtering could be one way of exploring this: <https://mr->

dictionary.mrcieu.ac.uk/term/steiger/#:~:text=Steiger filtering is a method,that the genetic variant influences.

>>Response: Thank you for your great suggestions. We agree that conducting Steiger filtering is a great way to investigating the directionality of MR results. According to your suggestion, we conducted MR-Steiger analysis, and showed that all the causal estimates were oriented in the intended direction (i.e., both forward and reverse direction. All $P_{MR-Steiger} < 0.05$). We have added description of MR-Steiger analysis in both Methods and Results section." *We also applied MR-Steiger with the R package "TwoSampleMR" to assure that the causal direction between the hypothesized exposure and outcome was correctly assigned*", (lines 217 to 219, page 11) and " *MR-Steiger results showed that all the causal estimates were oriented in the intended direction (All $P_{MR-Steiger} < 0.05$).*" (lines 414 to 415, page 20). Results for MR-Steiger were added in Supplementary Table 24.

It's good to consider the burden of multiple testing – however, you may wish to point out that some of your corrections may be overly stringent, given that the respiratory outcome you study are themselves highly correlated with one another (and your other traits will be correlated too).

>>Response: Thank you for your great suggestions. According to your suggestion, we have added one sentence in the Discussion section to make clear that some of our corrections may be overly stringent due to high correlation between traits we included. *"Fourthly, some of our analysis may suffer from multiple testing burden, because traits included in the present study, such as lung function parameters were highly correlated."*

Please give more details about the HEIDI method and how it works in terms of 'excluding pleiotropic effects'. My understanding is that it examines whether effect sizes for LD-correlated SNPs in a region are more different than one would expect by chance.

>>Response: Thank you for your suggestions. We now added a sentence in the Methods section to explain how HEIDI works in excluding pleiotropic effects. *"HEIDI is a method to detect pleiotropic SNPs at which the estimates of b_{xy} (the estimate of the effect of x on y) are significantly different from expected under a causal model, and remove them from the GSMR analysis, with accounting for LD among SNPs using the reference dataset for LD estimation.*

Results

The results section is very verbose with lots of long lists of locus names and numbers of loci. It feels fairly overwhelming to read at times, and I wonder if the section could be shortened, with just key messages highlighted (and the reader directed to figures and tables elsewhere).

>>Response: Thank you for your suggestions. We now cut down words from the Results section and moved some text of shared genes to the Discussion.

Observing shared loci in cross-trait meta-analysis is presumably to be expected – for example, FEV1 and FVC are highly correlated, so I am unsure how interesting it is that the same top SNP comes up for one of the loci mentioned.

>>Response: Thank you for your suggestions. We agree that it is not surprising to identified same shared loci of FEV1 and FVC with GlycA since FEV1 and FVC are highly correlated, it is therefore to certain extent highlight the role/pleiotropy effect of locus *chr19q13.32* in lung function decline or in linking inflammation to lung function decline (i.e., both FEV1 and FVC) since this locus related with these traits simultaneously.

Line 273 - novel candidate loci for COPD – please see Shrine et al 2023 which almost quadruples the known number of loci for lung function – are any of the loci mentioned in the paper?

>>Response: Thank you for your great suggestions. We compared our novel loci with the newest GWAS and five of them (*chr1p36.11*, *chr3p21.31*, *chr4p15.32*, *chr6p22.1*, and *chr10q21.3*) were also identified by the larger GWAS suggesting reliability of the cross-trait meta-analysis in increasing power of identifying novel genetic loci.

Line 291 – if loci are shared between GlycA and LF and CRP and LF, could it be that CRP is an upstream marker of GlycA?

>>Response: Thank you for your great suggestions. We agree that these overlapping signals may suggest shared pathway or potential interaction. According to your suggestion, we conducted pathway analysis using the shared genes of GlycA and CRP with lung function and found that glycosphingolipid biosynthesis and lipid glycosylation pathways may play a role. We added a sentence to discuss this finding in the Discussion section as suggested. *“Interestingly, some of these shared signals were also significant for other inflammatory markers, specifically CRP or albumin, which not only suggested that GlycA may capture the glycosylation states of CRP or albumin, but also indicated potential shared biological pathways, such as glycosphingolipid biosynthesis and lipid glycosylation pathways in relating inflammation to lung function decline and respiratory diseases.”* (lines 492 to 496, page 24)

I find it very surprising that the MR result was so totally null for COPD, given the strong MR results observed for asthma and lung function, and the strong genetic correlation with COPD. These traits are all highly genetically correlated, so to see no effect at all for COPD

strikes me as unusual. What explanations do the authors have for this? In what circumstances would you expect a null MR result (in both directions) in the present of a strong genetic correlation? It would be important to add this to the text.

>>Response: Thank you for your great suggestions. Significant genetic correlation (r_g estimated by LDSC) between traits is either mediated by shared etiology (horizontal pleiotropy) or by causation (vertical pleiotropy). Therefore, our results suggested that it is more likely that the observed associations between GlycA and COPD are explained by shared genetics and related shared etiology rather than causality. Another possible reason is that the number of instruments for COPD is much smaller than that of lung function. We added discussion accordingly: *“Additionally, as we observed strong genetic correlation but no significant instrumental effects between GlycA and COPD, it is more likely that the observed associations between GlycA and COPD are explained by shared genetics and related shared etiology rather than causality. Also, the number of instruments for COPD is much smaller than other traits or diseases and the reverse MR analysis might be underpowered.”*

Discussion

Can you comment on why you may have found only weak genetic correlation results between GlycA and FEV1/FVC? It seems odd, given the strong correlation with COPD. Moreover, the directions are in the opposite direction to what one might expect, given the association with COPD (higher values of FEV1/FVC associated with inflammation).

>>Response: Thank you for your great suggestions. A genetic correlation is the proportion of variance that two traits share due to genetic causes, the correlation between the genetic influences on a trait and the genetic influences on a different trait estimating the degree of pleiotropy or causal overlap. Therefore, the genetic correlation difference between FEV1/FVC and COPD might be the distribution difference of their significant genetic causes from GWAS (see details below for their GWAS Manhattan plot).

Can you comment on why you might have found no correlations with fibrinogen? Do you

think this is an artefact of smaller sample size or measurement error, or might there be a biological reason?

>>Response: Thank you for your great suggestions. The most possible reason is that the sample size is relatively smaller for fibrinogen, future studies with larger sample size are warranted to further explore the genetic association between fibrinogen, lung function and respiratory diseases. We have added a sentence to discuss this in the Discussion section. *" Additionally, the null results of fibrinogen in our study might be a consequence of smaller sample size of fibrinogen GWAS."*

As mentioned above, if GlycA is predictive of T2D, CVD AND respiratory disease, it doesn't strike me as a particularly useful predictive marker, as it is non-specific. I can however see that it might be useful in flagging a harmful pro-inflammatory state. If it is indeed true that raised GlycA levels predict and precede disease (and the reverse MR result does make me question this somewhat), what mechanisms do you hypothesise for its elevation? Could it be a marker of smoking, for example?

>>Response: Thank you for your great suggestions. We agree that GlycA is a composite measure of inflammation related with multiple chronic diseases and the solely risk prediction value for lung function and respiratory diseases warrant further investigation. However, previous studies suggested that a combination of low-grade systemic inflammation markers yields more information than the biomarkers assessed individually (PMID: 22427534, 10733371, 30170809). Therefore, since GlycA reflects multiple inflammatory pathways and chronic inflammation status, and may serve as a complementary marker to other inflammatory markers, adding GlycA into a risk prediction model could potentially increase the prediction accuracy. According to your suggestion, we added one sentence to discuss the clinical utility of GlycA. *" However, GlycA is a composite measure of inflammation, the clinical utility as an inflammation biomarker or its potential in translating to novel therapies for lung function and respiratory diseases warrant further investigation"* (lines 448 to 450, page 22)

I am unsure about how GlycA might translate to novel clinical therapies, if it is an end-marker of multiple inflammatory pathways. You acknowledge this in the limitations, but still mention it earlier on in the discussion. Similarly, if ubiquitination is indeed a 'ubiquitous' process in the body, is it likely to be a good drug target? I was interested to see this paper, which suggests it might be possible, although I imagine it would be very fraught with difficulty <https://www.nature.com/articles/cr201631>

>>Response: Thank you for your great suggestions. We agree that there is still uncertainty by saying the shared signals may translate into novel clinical therapies based on our results. We have tuned down our statement across our manuscript. As for ubiquitination, we rewrote the sentence describing UBA7. *"TWAS results further suggested that UBA7 gene was a potential candidate gene for these traits within this locus, suggesting the signaling pathways regulated by ubiquitination may play roles in lung function decline. Ubiquitination was reported to play an*

important role in the pathobiology of lung injury as it regulates the proteins modulating the alveolocapillary barrier and the inflammatory response.”

REVIEWER COMMENTS

Reviewer #2 (Remarks to the Author):

Thank you to the authors for their revised manuscript; I know it is laborious to prepare revisions, and I do find the paper improved. I remain concerned about the utility of GlycA as either a drug target or biomarker, given it is an end point of multiple processes, and predictive of multiple chronic diseases. Clearly this is more of a conceptual concern, and not one that can be resolved by revising the paper, although it is helpful that the authors have added to the Discussion. Personally, I would probably add more strong caveats still, but other reviewers and editors may think differently.

Understanding the role of inflammation as cause (and /or) consequence or marker of lung function is still an interesting question, and one that this paper seeks to address. I think there is reasonable evidence as presented that the reverse direction (LF → GlycA) could still be causal as well (e.g. there is bidirectionality in effect). Perhaps you could remove SNP IVs that explain more variance in GlycA than LF and repeat the reverse MR to try and test this? See below comment on Steiger filtering.

Re: Figure 1 and my previous comments. Please note my comment responding to the information you provide in Manhattan plots re: the possible low genetic correlation between FEV1/FVC and COPD results. This strikes me as very surprising, since COPD in epidemiological studies is generally a dichotomisation of FEV1/FVC. Previous studies have indicated a genetic correlation of -0.84 (e.g, very, very strong) between COPD and FEV1/FVC, using independent GWAS. Unless I am missing something (possible), surely you would expect to see i) a substantial genetic correlation between the two traits here, and b) therefore, a negative association between GlycA and FEV1/FVC in Figure 1, given that you see a positive genetic correlation between GlycA and COPD ? Results from the FEV1/FVC analysis should be better powered than the COPD results, so it seems strange to me that you do not observe anything strong at all (unsure what exact p-value for GlycA FEV1/FVC correlation is, but there are no significance 'stars **', and the effect direction showsn (0.06) is not logically consistent with COPD risk (0.31), as effects on FEV1/FVC and COPD should be opposing – Biologically, higher ratio means indicates lower COPD risk?).

I have the following remaining concerns:

- Abstract – why are your results less likely to be affected by selection bias, just because you have used large studies? Cohorts are notorious for being affected by selection bias.
- You talk about lung function decline throughout, but haven't you only looked at GWAS of cross-sectionally measured lung function? If you have used cross-sectional GWAS only, this is not studying lung function decline.
- See also above. I am aware that you now cite this paper (<https://link.springer.com/article/10.1007/s10654-021-00726-8>), I did a ctrl+F for 'Steiger' in your supplement (427952_1_supp_8388198_s54xtc.pdf) and cannot find any added result as you mention in your response letter. Some tables are truncated at the right edge for me though, as the pages are all in portrait view – maybe it is a format issue. Also, what is the PMR-STEIGER that you refer to in the Results section of the main manuscript? Is this testing that the variance explained by the SNPs in the outcome is

less than that explained in the exposure? If so, I would be more explicit about this. I would be tempted to apply Steiger 'filtering' (i.e. dropping IVs that explain more variance in GlycA than LF) to the reverse direction MR. If you see no evidence of association after doing so, this would help you be more confident in the forward GlycA↔LF association. As your results stand currently, you do observe strong effects in the reverse direction, so if the reverse direction MR withstands Steiger filtering, I would stress the possibility of bidirectionality more strongly.

- Re: [[Response: Thank you for your great suggestions. According to your suggestion, we have added one sentence in the Discussion section to make clear that some of our corrections may be overly stringent due to high correlation between traits we included. "Fourthly, some of our analysis may suffer from multiple testing burden, because traits included in the present study, such as lung function parameters were highly correlated."]]

- I would edit to say 'some of our multiple testing corrections may be overly stringent'

REVIEWER COMMENTS

Reviewer #2 (Remarks to the Author):

1. Thank you to the authors for their revised manuscript; I know it is laborious to prepare revisions, and I do find the paper improved. I remain concerned about the utility of GlycA as either a drug target or biomarker, given it is an end point of multiple processes, and predictive of multiple chronic diseases. Clearly this is more of a conceptual concern, and not one that can be resolved by revising the paper, although it is helpful that the authors have added to the Discussion. Personally, I would probably add more strong caveats still, but other reviewers and editors may think differently. >>Response: Thank you for your positive remarks, valuable suggestions, and comments to help us improve our manuscript.

We agree the utility of GlycA as either a drug target or biomarker is of great importance, since identifying biomarkers reliably reflecting systemic, chronic and low-grade inflammation is a key challenge for inflammation research. Recent studies suggested that given GlycA is a composite marker of inflammation, it should be less reactive to acute environmental changes and, as such, be a more stable measure of chronic inflammation compared to acute-phase markers such as most widely used biomarker-hsCRP (PMID: 29078787).

Interestingly, Crick et al. (PMID: 38148627, online on 26 December 2023) recently conducted a study leveraging two large UK population-based cohorts: the Avon Longitudinal Study of Parents and Children (ALSPAC) and the UK Biobank (UKB) to investigate the stability of GlycA: (1) short-term (weeks) and long-term (years) stability of GlycA levels; (2) correlations between concurrently measured GlycA and hsCRP in adolescence, early-adulthood and mid-adulthood; (3) associations of inflammation-related factors (e.g., smoking) with GlycA and hsCRP and (4) associations of autoimmune/inflammatory diseases with GlycA and hsCRP. Their results suggested that GlycA showed greater stability over time than hsCRP and display similar or stronger associations with inflammation-related factors (particularly chronic inflammatory states) compared to hsCRP. We have added this in the Discussion Section (lines 430-437). *"GlycA is a composite marker of inflammation, it should be less reactive to acute environmental changes and, therefore, be a more stable measure of chronic inflammation compared to acute-phase markers such as most widely used biomarker-hsCRP. Recently, Crick et al. conducted a study leveraging two large UK population-based cohorts: the Avon Longitudinal Study of Parents and Children (ALSPAC) and the UKB to investigate the stability of GlycA, and their results suggested that GlycA showed greater stability over time than hsCRP and displayed similar or stronger associations with inflammation-related factors (particularly chronic inflammatory states) compared to hsCRP"*

2. Understanding the role of inflammation as cause (and /or) consequence or marker of lung function is still an interesting question, and one that this paper seeks to address. I think there is reasonable evidence as presented that the reverse direction (LF \diamond GlycA) could still be causal as well (e.g. there is bidirectionality in effect). Perhaps you could remove SNP IVs that explain more variance in GlycA than LF and repeat the reverse MR to try and test this? See below comment on Steiger filtering.

>>Response: Thank you for your comments. We conducted Steiger directionality test in the previous revision and included p-values for Steiger directionality test in Supplementary table 24 (all p values were small suggesting that the overall direction of the observed MR associations was correct). Then, we conducted Steiger filtering to check if any of the instrument variables explained substantially more variance in the exposure than in the outcome (we now added the results in the Results Section and Supplementary files). We found all SNPs in the forward direction survived Steiger filtering. In the reverse direction, we filtered out 11 SNPs (rs149717632, rs17427599, rs2858863, rs3135387, rs330925, rs342466, rs550057, rs56083805, rs62053264, rs62091797, rs695238), 13 SNPs (rs10737680, rs12132138, rs1299113, rs2927325, rs3130283, rs56349217, rs62091797, rs7092539, rs72738786, rs7613360, rs7947322, rs8126001, rs9411378), 9 SNPs (rs113638840, rs11759008, rs2414870, rs4987082, rs60648773, rs73390208, rs7896518, rs9264803, rs983309) and 7 SNPs (rs2260000, rs2352974, rs2571392, rs2844498, rs529565, rs56121115, rs713916, we only included p values for overall Steiger directionality test in the table to minimize the space for supplementary files) for FEV1, FVC, FEV1/FVC and PEF, respectively, but not asthma and COPD. We then repeated the reverse MR after Steiger filtering, our results showed consistent results with the primary GSMR and MR-PRESSO results before filtering. we now revised the sentences accordingly in the manuscript (see details in the response below on comments of Steiger filtering).

3. Re: Figure 1 and my previous comments. Please note my comment responding to the information you provide in Manhattan plots re: the possible low genetic correlation between FEV1/FVC and COPD results. This strikes me as very surprising, since COPD in epidemiological studies is generally a dichotomisation of FEV1/FVC. Previous studies have indicated a genetic correlation of -0.84 (e.g, very, very strong) between COPD and FEV1/FVC, using independent GWAS. Unless I am missing something (possible), surely you would expect to see i) a substantial genetic correlation between the two traits here, and b) therefore, a negative association between GlycA and FEV1/FVC in Figure 1, given that you see a positive genetic correlation between GlycA and COPD ? Results from the FEV1/FVC analysis should be better powered than the COPD results, so it seems strange to me that you do not observe anything strong at all (unsure what exact p-value for GlycA FEV1/FVC correlation is, but there are no significance 'stars **', and the effect direction shown (0.06) is not logically consistent with COPD risk (0.31), as effects on FEV1/FVC and COPD

should be opposing - Biologically, higher ratio means indicates lower COPD risk?).

>>Response: Thank you for your comments. We agree that COPD is diagnosed by spirometric decreases in lung function, namely forced expiratory volume in one second (FEV1) and its ratio to forced vital capacity (FEV1/FVC) (PMID: 33057025). According to previous studies, COPD has substantial heritability, with estimates ranging from 35–60%. While lung function parameters are also heritable in the general population with over 40% of variation in FEV1, FVC and FEV1/FVC attributable to genetic factors (PMID: 11316667).

In the current study, we used GWAS summary-level data for four lung function parameters by Shrine et al. published on 2019 using UK Biobank data (PMID: 30804560). According to UK Biobank, the data on breath spirometry was a single measurement collected using a Vitalograph Pneumotrac 6800. The participant is asked to record two to three blows (lasting for at least 6 seconds) within a period of about 6 minutes. The computer compares the reproducibility of the first two blows and, if acceptable (defined as a <5% difference in forced volume vital capacity (FVC) & Forced Expiratory Volume in 1 second (FEV1)), will indicate that the third blow is not required. The variations for FEV1/FVC in the UK Biobank are relatively small (ranges from 0.64 to 0.80, see below for details in the figure 1 and 2), and the original GWAS lack information on medications. We used GWAS summary-level data for asthma and COPD from the National Bioscience Database Center (NBDC) human database (<https://humandbs.biosciencedbc.jp/en/>) (PMID: 34594039) using ICD code: J44.

We now conducted genetic correlation between COPD/asthma and lung function parameters, and their genetic correlation ranges from -0.22 to -0.47 in the current dataset from general population (see table 1 below for details. Therefore, it is possible the previous study of -0.84 is based on family-based dataset). In the recent lung function GWAS (PMID: 36914875), although Shrine N et al didn't investigated the genetic correlation between lung function and COPD, they did build a FEV1/FVC GRS and investigated its association with COPD showing a odds ratio for COPD per s.d. of GRS increase was 1.63 (i.e., decrement in genetic determined FEV1/FVC was related with 63% increased risk of COPD, which could translate into a β coefficient of 0.49, consistent in the magnitude and direction of our genetic correlation analysis of -0.42, i.e., increment in genetic determined FEV1/FVC was related with decreased risk of COPD).

As for the magnitude and direction of genetic correlation of GlycA with FEV1/FVC, the genetic correlation between GlycA and FEV1/FVC is not significant in our analysis after correcting for multiple testing, which is consistent in our sensitivity analysis using an independent dataset from SpiroMeta consortium (see below for details in Table2). The reasons for this might be 1) the power differences in these two GWASs; 2) LD score regression used HAPMAP3 SNPs and might be less powerful when genetic correlation is moderate (which is the case in the association of glyca

with lung function parameters); 3) lacking information on medications in the original GWASs; 4) differences in heritability of these two traits (PMID: 29220677, 25642630). We have added this in the discussion to further explain the genetic association of GlycA with FEV1/FVC. *“Additionally, despite the strong phenotypic association of FEV1/FVC with COPD in observational studies, we observed significant genetic correlation between glycA and COPD but null for FEV1/FVC. The possible reason for the differences in the magnitude of genetic correlation might be explained by differences in heritability, lack of information on medications, power differences of these two GWASs, and less power for LD score regression when genetic correlation is moderate. Although sensitivity analysis using independent dataset from SpiroMeta consortium showed consistent results, future studies with larger sample size, repeated measurements and information on respiratory medications are warranted to validate the associations of FEV1/FVC and COPD with glycA.”* (lines 440 to 448, page 21).

Figure 1 the distribution of FEV1/FVC according to age group in UK Biobank

Figure 2 the frequency distribution of FEV1/FVC in UK Biobank

Table 1 genetic correlation of COPD and asthma with lung function

Trait1	Trait2	rg	se	z	p
COPD	FEV1_meta-analysis	-0.47	0.03	-17.09	1.80E-65
	FVC_meta-analysis	-0.28	0.03	-10.51	7.46E-26
	FEV1_to_FVC_RATIO_meta-analysis	-0.42	0.03	-14.15	1.84E-45
	PEF_meta-analysis	-0.38	0.03	-13.16	1.41E-39
	UKBiobank_FEV1	-0.45	0.03	-16.01	1.11E-57
	UKBiobank_FVC	-0.27	0.03	-9.30	1.36E-20
	UKBiobank_FEV1_to_FVC_RATIO	-0.41	0.03	-14.05	7.57E-45
Asthma	UKBiobank_PEF	-0.37	0.03	-13.51	1.29E-41
	FEV1_meta-analysis	-0.38	0.02	-15.39	1.81E-53
	FVC_meta-analysis	-0.22	0.03	-8.22	2.01E-16
	FEV1_to_FVC_RATIO_meta-analysis	-0.35	0.03	-13.16	1.55E-39
	PEF_meta-analysis	-0.30	0.03	-11.59	4.44E-31
	UKBiobank_FEV1	-0.37	0.03	-14.82	1.15E-49
	UKBiobank_FVC	-0.22	0.03	-7.97	1.54E-15
UKBiobank_FEV1_to_FVC_RATIO	-0.34	0.03	-13.14	2.01E-39	

Table 2 sensitivity analysis of genetic correlation of inflammation biomarkers with lung function using independent dataset.

Trait1	Trait2	rg	se	z	p
GlycA	FVC	-0.20	0.03	-5.93	3.08E-09
	FEV1	-0.17	0.04	-4.02	5.84E-05
	FEV1_to_FVC_RATIO	0.03	0.05	0.55	0.58
	PEF	-0.13	0.08	-1.73	0.08
CRP	FVC	-0.26	0.03	-8.35	6.97E-17
	FEV1	-0.25	0.03	-8.34	7.77E-17
	FEV1_to_FVC_RATIO	-0.02	0.03	-0.46	0.65
	PEF	-0.13	0.05	-2.46	0.01
Fibrinogen	FVC	0.04	0.09	0.38	0.70
	FEV1	0.00	0.10	0.03	0.97
	FEV1_to_FVC_RATIO	0.00	0.14	0.00	1.00
	PEF	-0.28	0.16	-1.72	0.08
Albumin	FVC	0.05	0.03	1.51	0.13
	FEV1	0.07	0.03	2.38	0.02
	FEV1_to_FVC_RATIO	0.03	0.03	0.99	0.32
	PEF	-0.04	0.05	-0.83	0.41
WBC	FVC	-0.06	0.03	-2.08	0.04
	FEV1	-0.09	0.03	-2.99	2.81E-03
	FEV1_to_FVC_RATIO	-0.06	0.04	-1.61	0.11
	PEF	-0.14	0.05	-2.64	0.01

4. I have the following remaining concerns:

- Abstract - why are your results less likely to be affected by selection bias, just because you have used large studies? Cohorts are notorious for being affected by selection bias.

>>Response: Thank you for your comments. Whenever possible, it is preferable to obtain all relevant genotype data and correct for confounding biases directly. According to previous studies, both polygenicity (*i.e.*, many small genetic effects) and confounding biases, such as cryptic relatedness and population stratification, can yield an inflated distribution of test statistics in genome-wide association studies (GWAS). However, in the event that only summary data are available, or if a conservative correction is desired, LD Score regression intercept provides a more robust quantification of the extent of inflation from confounding bias than λ_{GC} . Since λ_{GC} increases with sample size in the presence of polygenicity (even without confounding bias), the gain in power obtained by correcting test statistics with the LD Score regression intercept instead of λ_{GC} will become even more substantial for larger GWAS. Therefore, LD Score regression provides a method less likely to be biased from confounding bias and polygenicity as compared to conventional observational studies (*i.e.*, phenotypic analysis, details for this method can be found

in the original paper: Bulik-Sullivan B, Finucane HK et al., An atlas of genetic correlations across human diseases and traits, Nat Genet, 2015, PMID: 26414676). We have revised it accordingly in the Abstract. *“All analyses were less likely to be affected by reverse causation and selection bias as well as environmental and behavioral confounding as compared to conventional observational studies.”*

5. - You talk about lung function decline throughout, but haven't you only looked at GWAS of cross-sectionally measured lung function? If you have used cross-sectional GWAS only, this is not studying lung function decline.

>>Response: Thank you for your suggestions, we have now revised as “lung function” throughout the manuscript.

6. - See also above. I am aware that you now cite this paper (<https://link.springer.com/article/10.1007/s10654-021-00726-8>), I did a ctrl+F for ‘Steiger’ in your supplement (427952_1_supp_8388198_s54xtc.pdf) and cannot find any added result as you mention in your response letter. Some tables are truncated at the right edge for me though, as the pages are all in portrait view - maybe it is a format issue. Also, what is the PMR-STEIGER that you refer to in the Results section of the main manuscript? Is this testing that the variance explained by the SNPs in the outcome is less than that explained in the exposure? If so, I would be more explicit about this. I would be tempted to apply Steiger ‘filtering’ (i.e. dropping IVs that explain more variance in GlycA than LF) to the reverse direction MR. If you see no evidence of association after doing so, this would help you be more confident in the forward GlycA Δ LF association. As your results stand currently, you do observe strong effects in the reverse direction, so if the reverse direction MR withstands Steiger filtering, I would stress the possibility of bidirectionality more strongly.

>>Response: Thank you for your comments. We are sorry for the format issue in the supplementary files, we have gone through the supplementary files to make sure all the tables are in correct format.

We conducted Steiger directionality test in the previous revision and included p-values for Steiger directionality test in Supplementary table 24 (all p values were small suggesting that the overall direction of the observed MR associations was correct). Then, we conducted Steiger filtering to check if any of the instrument variables explained substantially more variance in the exposure than in the outcome, we found all SNPs in the forward direction survived Steiger filtering (we now added the related results in the Results Section and Supplementary files). In the reverse direction, we filtered out 11 SNPs (rs149717632, rs17427599, rs2858863, rs3135387,

rs330925, rs342466, rs550057, rs56083805, rs62053264, rs62091797, rs695238), 13 SNPs (rs10737680, rs12132138, rs1299113, rs2927325, rs3130283, rs56349217, rs62091797, rs7092539, rs72738786, rs7613360, rs7947322, rs8126001, rs9411378), 9 SNPs (rs113638840, rs11759008, rs2414870, rs4987082, rs60648773, rs73390208, rs7896518, rs9264803, rs983309) and 7 SNPs (rs2260000, rs2352974, rs2571392, rs2844498, rs529565, rs56121115, rs713916, we only included p values for overall Steiger directionality test in the table to minimize the space for supplementary files) for FEV1, FVC, FEV1/FVC and PEF, respectively and repeated the reverse MR, our results showed consistent results with the primary GSMR and MR-PRESSO results before filtering (see table below for the results after Steiger filtering). This is because our primary MR analysis using GSMR have filtered pleiotropic SNPs using HEIDI. HEIDI is a stringent method to detect pleiotropic SNPs at which the estimates of bxy (the estimate of the effect of x on y) are significantly different from expected under a causal model, and remove them from the GSMR analysis, with accounting for LD among SNPs using the reference dataset for LD estimation. We mainly observed consistent significant instrumental effect of FEV1 and FVC with glycA in the reverse direction. As suggested, we have revised the sentences accordingly in the manuscript. " Additionally, sensitivity analysis using MR_PRESSO showed consistent results with GSMR, and Steiger directionality test showed that all the causal estimates were oriented in the intended direction (i.e., all $P_{MR-Steiger} < 0.05$ suggesting that the overall direction of the observed MR associations were correct, Supplementary Table 24). Furthermore, MR results after Steiger filtering showed consistent results with GSMR and suggested the association of glycA with FEV1 and FVC are more likely to be bidirectional (Supplementary Table 25)." (lines 395 to 401, page 19)

Exposure	Outcome	Causal Estimate	Sd	T-stat	P-value
FEV1	glycA	-0.07	0.01	-4.90	1.31E-06
FVC	glycA	-0.10	0.02	-5.78	1.52E-08
FEV1_FVC_Ratio	glycA	0.05	0.01	4.66	3.98E-06
PEF	glycA	-0.01	0.01	-0.31	0.76
FEV1	CRP	0.00	0.00	-0.43	0.67
FVC	CRP	-0.02	0.01	-3.04	2.57E-03
FEV1_FVC_Ratio	CRP	0.00	0.00	0.34	0.73
PEF	CRP	0.00	0.01	-0.21	0.83
FEV1	albumin	0.00	0.01	-0.05	0.96
FVC	albumin	0.01	0.01	0.50	0.62
FEV1_FVC_Ratio	albumin	-0.01	0.01	-1.07	0.28
PEF	albumin	-0.01	0.01	-1.33	0.19

7. - Re: [[Response: Thank you for your great suggestions. According to your suggestion, we have added one sentence in the Discussion section to make clear that some of our corrections may be overly stringent due to high correlation between traits we included. “Fourthly, some of our analysis may suffer from multiple testing burden, because traits included in the present study, such as lung function parameters were highly correlated.”]]

- I would edit to say ‘some of our multiple testing corrections may be overly stringent’

>>Response: Thank you for your suggestion. We have revised it as “Fourthly, some of our multiple testing corrections may be overly stringent...” .

REVIEWERS' COMMENTS

Reviewer #2 (Remarks to the Author):

Please see attached MS Word document.

REVIEWER COMMENTS

Reviewer #2 (Remarks to the Author):

1. Thank you to the authors for their revised manuscript; I know it is laborious to prepare revisions, and I do find the paper improved. I remain concerned about the utility of GlycA as either a drug target or biomarker, given it is an end point of multiple processes, and predictive of multiple chronic diseases. Clearly this is more of a conceptual concern, and not one that can be resolved by revising the paper, although it is helpful that the authors have added to the Discussion. Personally, I would probably add more strong caveats still, but other reviewers and editors may think differently. >>Response: Thank you for your positive remarks, valuable suggestions, and comments to help us improve our manuscript.

We agree the utility of GlycA as either a drug target or biomarker is of great importance, since identifying biomarkers reliably reflecting systemic, chronic and low-grade inflammation is a key challenge for inflammation research. Recent studies suggested that given GlycA is a composite marker of inflammation, it should be less reactive to acute environmental changes and, as such, be a more stable measure of chronic inflammation compared to acute-phase markers such as most widely used biomarker-hsCRP (PMID: 29078787).

Interestingly, Crick et al. (PMID: 38148627, online on 26 December 2023) recently conducted a study leveraging two large UK population-based cohorts: the Avon Longitudinal Study of Parents and Children (ALSPAC) and the UK Biobank (UKB) to investigate the stability of GlycA: (1) short-term (weeks) and long-term (years) stability of GlycA levels; (2) correlations between concurrently measured GlycA and hsCRP in adolescence, early-adulthood and mid-adulthood; (3) associations of inflammation-related factors (e.g., smoking) with GlycA and hsCRP and (4) associations of autoimmune/inflammatory diseases with GlycA and hsCRP. Their results suggested that GlycA showed greater stability over time than hsCRP and display similar or stronger associations with inflammation-related factors (particularly chronic inflammatory states) compared to hsCRP. We have added this in the Discussion Section (lines 430-437). *“GlycA is a composite marker of inflammation, it should be less reactive to acute environmental changes and, therefore, be a more stable measure of chronic inflammation compared to acute-phase markers such as most widely used biomarker-hsCRP. Recently, Crick et al. conducted a study leveraging two large UK population-based cohorts: the Avon Longitudinal Study of Parents and Children (ALSPAC) and the UKB to investigate the stability of GlycA, and their results suggested that GlycA showed greater stability over time than hsCRP and displayed similar or stronger associations with inflammation-related factors (particularly chronic inflammatory states) compared to hsCRP”*

Thank you for your response. My concern was not whether GlycA is a true biomarker for inflammation, but more how specific it would be in predicting respiratory disease versus many other disease (even ignoring any association in the reverse direction).

2. Understanding the role of inflammation as cause (and /or) consequence or marker of lung function is still an interesting question, and one that this paper seeks to address. I think there is reasonable evidence as presented that the reverse direction (LF \diamond GlycA) could still be causal as well (e.g. there is bidirectionality in effect). Perhaps you could remove SNP IVs that explain more variance in GlycA than LF and repeat the reverse MR to try and test this? See below comment on Steiger filtering.

>>Response: Thank you for your comments. We conducted Steiger directionality test in the previous revision and included p-values for Steiger directionality test in Supplementary table 24 (all p values were small suggesting that the overall direction of the observed MR associations was correct). Then, we conducted Steiger filtering to check if any of the instrument variables explained substantially more variance in the exposure than in the outcome (we now added the results in the Results Section and Supplementary files). We found all SNPs in the forward direction survived Steiger filtering. In the reverse direction, we filtered out 11 SNPs (rs149717632, rs17427599, rs2858863, rs3135387, rs330925, rs342466, rs550057, rs56083805, rs62053264, rs62091797, rs695238), 13 SNPs (rs10737680, rs12132138, rs1299113, rs2927325, rs3130283, rs56349217, rs62091797, rs7092539, rs72738786, rs7613360, rs7947322, rs8126001, rs9411378), 9 SNPs (rs113638840, rs11759008, rs2414870, rs4987082, rs60648773, rs73390208, rs7896518, rs9264803, rs983309) and 7 SNPs (rs2260000, rs2352974, rs2571392, rs2844498, rs529565, rs56121115, rs713916, we only included p values for overall Steiger directionality test in the table to minimize the space for supplementary files) for FEV1, FVC, FEV1/FVC and PEF, respectively, but not asthma and COPD. We then repeated the reverse MR after Steiger filtering, our results showed consistent results with the primary GSMR and MR-PRESSO results before filtering. we now revised the sentences accordingly in the manuscript (see details in the response below on comments of Steiger filtering).

Thank you – so as you state, you do find evidence of bidirectionality in your analyses (e.g. GlycA may cause lung function, but equally, lung function may change GlycA). I would have preferred that you had made more of this in the Discussion, but I will leave it to editorial discretion as to whether that is appropriate.

3. Re: Figure 1 and my previous comments. Please note my comment responding to the information you provide in Manhattan plots re: the possible low genetic correlation between FEV1/FVC and COPD results. This strikes me as very surprising, since COPD in epidemiological studies is generally a dichotomisation of FEV1/FVC. Previous studies have indicated a genetic correlation of -0.84 (e.g, very, very strong) between COPD and FEV1/FVC, using independent GWAS. Unless I am missing something (possible), surely you would

expect to see i) a substantial genetic correlation between the two traits here, and b) therefore, a negative association between GlycA and FEV1/FVC in Figure 1, given that you see a positive genetic correlation between GlycA and COPD ? Results from the FEV1/FVC analysis should be better powered than the COPD results, so it seems strange to me that you do not observe anything strong at all (unsure what exact p-value for GlycA FEV1/FVC correlation is, but there are no significance ‘stars **’, and the effect direction shown (0.06) is not logically consistent with COPD risk (0.31), as effects on FEV1/FVC and COPD should be opposing - Biologically, higher ratio means indicates lower COPD risk?).

>>Response: Thank you for your comments. We agree that COPD is diagnosed by spirometric decreases in lung function, namely forced expiratory volume in one second (FEV1) and its ratio to forced vital capacity (FEV1/FVC) (PMID: 33057025). According to previous studies, COPD has substantial heritability, with estimates ranging from 35–60%. While lung function parameters are also heritable in the general population with over 40% of variation in FEV1, FVC and FEV1/FVC attributable to genetic factors (PMID: 11316667).

In the current study, we used GWAS summary-level data for four lung function parameters by Shrine et al. published on 2019 using UK Biobank data (PMID: 30804560). According to UK Biobank, the data on breath spirometry was a single measurement collected using a Vitalograph Pneumotrac 6800. The participant is asked to record two to three blows (lasting for at least 6 seconds) within a period of about 6 minutes. The computer compares the reproducibility of the first two blows and, if acceptable (defined as a <5% difference in forced volume vital capacity (FVC) & Forced Expiratory Volume in 1 second (FEV1)), will indicate that the third blow is not required. The variations for FEV1/FVC in the UK Biobank are relatively small (ranges from 0.64 to 0.80, see below for details in the figure 1 and 2), and the original GWAS lack information on medications. We used GWAS summary-level data for asthma and COPD from the National Bioscience Database Center (NBDC) human database (<https://humandbs.biosciencedbc.jp/en/>) (PMID: 34594039) using ICD code: J44.

Thank you for clarifying.

We now conducted genetic correlation between COPD/asthma and lung function parameters, and their genetic correlation ranges from -0.22 to -0.47 in the current dataset from general population (see table 1 below for details. Therefore, it is possible the previous study of -0.84 is based on family-based dataset).

The value of -0.84 is taken from a genetic correlation of the Shrine et al 2019 FEV1/FVC data, and the Sakornsakolpat et al 2019 COPD data. But it is nevertheless reassuring that you see a correlation between COPD and FEV1/FVC.

In the recent lung function GWAS (PMID: 36914875), although Shrine N et al didn't investigated the genetic correlation between lung function and COPD, they did

build a FEV1/FVC GRS and investigated its association with COPD showing a odds ratio for COPD per s.d. of GRS increase was 1.63 (i.e., decrement in genetic determined FEV1/FVC was related with 63% increased risk of COPD, which could translate into a β coefficient of 0.49, consistent in the magnitude and direction of our genetic correlation analysis of -0.42, i.e., increment in genetic determined FEV1/FVC was related with decreased risk of COPD).

I agree – the correlation you observe is entirely consistent with that observed in Shrine et al 2019 – it is helpful to know that.

As for the magnitude and direction of genetic correlation of GlycA with FEV1/FVC, the genetic correlation between GlycA and FEV1/FVC is not significant in our analysis after correcting for multiple testing, which is consistent in our sensitivity analysis using an independent dataset from SpiroMeta consortium (see below for details in Table2). The reasons for this might be 1) the power differences in these two GWASs; 2) LD score regression used HAPMAP3 SNPs and might be less powerful when genetic correlation is moderate (which is the case in the association of glycA with lung function parameters); 3) lacking information on medications in the original GWASs; 4) differences in heritability of these two traits (PMID: 29220677, 25642630). We have added this in the discussion to further explain the genetic association of GlycA with FEV1/FVC. *“Additionally, despite the strong phenotypic association of FEV1/FVC with COPD in observational studies, we observed significant genetic correlation between glycA and COPD but null for FEV1/FVC. The possible reason for the differences in the magnitude of genetic correlation might be explained by differences in heritability, lack of information on medications, power differences of these two GWASs, and less power for LD score regression when genetic correlation is moderate. Although sensitivity analysis using independent dataset from SpiroMeta consortium showed consistent results, future studies with larger sample size, repeated measurements and information on respiratory medications are warranted to validate the associations of FEV1/FVC and COPD with glycA.”* (lines 440 to 448, page 21).

Thank you for all of this extra work. I remain struck by the fact that you see a more significant genetic correlation between GlycA and COPD than you do with GlycA and FEV1/FVC, because I would expect the power to be greater for the FEV1/FVC GWAS (as this is a quantitative trait), unless the sample size for the COPD GWAS is enormous I wonder if it's possible that the COPD GWAS is capturing something else, e.g. smoking, which would be associated with inflammation, and that this is driving the correlation (you may have thoughts on this)? Anyway, I think all you can do is (as you have done), be frank about what you have found.

Figure 1 the distribution of FEV1/FVC according to age group in UK Biobank

Figure 2 the frequency distribution of FEV1/FVC in UK Biobank

Table 1 genetic correlation of COPD and asthma with lung function

Trait1	Trait2	rg	se	z	p
COPD	FEV1_meta-analysis	-0.47	0.03	-17.09	1.80E-65
	FVC_meta-analysis	-0.28	0.03	-10.51	7.46E-26
	FEV1_to_FVC_RATIO_meta-analysis	-0.42	0.03	-14.15	1.84E-45

	PEF_meta-analysis	-0.38	0.03	-13.16	1.41E-39
	UKBiobank_FEV1	-0.45	0.03	-16.01	1.11E-57
	UKBiobank_FVC	-0.27	0.03	-9.30	1.36E-20
	UKBiobank_FEV1_to_FVC_RATIO	-0.41	0.03	-14.05	7.57E-45
	UKBiobank_PEF	-0.37	0.03	-13.51	1.29E-41
	FEV1_meta-analysis	-0.38	0.02	-15.39	1.81E-53
	FVC_meta-analysis	-0.22	0.03	-8.22	2.01E-16
	FEV1_to_FVC_RATIO_meta-analysis	-0.35	0.03	-13.16	1.55E-39
Asthma	PEF_meta-analysis	-0.30	0.03	-11.59	4.44E-31
	UKBiobank_FEV1	-0.37	0.03	-14.82	1.15E-49
	UKBiobank_FVC	-0.22	0.03	-7.97	1.54E-15
	UKBiobank_FEV1_to_FVC_RATIO	-0.34	0.03	-13.14	2.01E-39
	UKBiobank_PEF	-0.29	0.02	-11.71	1.13E-31

Table 2 sensitivity analysis of genetic correlation of inflammation biomarkers with lung function using independent dataset.

Trait1	Trait2	rg	se	z	p
	FVC	-0.20	0.03	-5.93	3.08E-09
	FEV1	-0.17	0.04	-4.02	5.84E-05
GlycA	FEV1_to_FVC_RATIO	0.03	0.05	0.55	0.58
	PEF	-0.13	0.08	-1.73	0.08
	FVC	-0.26	0.03	-8.35	6.97E-17
	FEV1	-0.25	0.03	-8.34	7.77E-17
CRP	FEV1_to_FVC_RATIO	-0.02	0.03	-0.46	0.65
	PEF	-0.13	0.05	-2.46	0.01
	FVC	0.04	0.09	0.38	0.70
	FEV1	0.00	0.10	0.03	0.97
Fibrinogen	FEV1_to_FVC_RATIO	0.00	0.14	0.00	1.00
	PEF	-0.28	0.16	-1.72	0.08
	FVC	0.05	0.03	1.51	0.13
	FEV1	0.07	0.03	2.38	0.02
Albumin	FEV1_to_FVC_RATIO	0.03	0.03	0.99	0.32
	PEF	-0.04	0.05	-0.83	0.41
	FVC	-0.06	0.03	-2.08	0.04
	FEV1	-0.09	0.03	-2.99	2.81E-03
WBC	FEV1_to_FVC_RATIO	-0.06	0.04	-1.61	0.11
	PEF	-0.14	0.05	-2.64	0.01

4. I have the following remaining concerns:

- Abstract - why are your results less likely to be affected by selection bias, just because you have used large studies? Cohorts are notorious for being affected by selection bias.

>>Response: Thank you for your comments. Whenever possible, it is preferable to obtain all relevant genotype data and correct for confounding biases directly. According to previous studies, both polygenicity (*i.e.*, many small genetic effects) and confounding biases, such as cryptic relatedness and population stratification, can yield an inflated distribution of test statistics in genome-wide association studies (GWAS). However, in the event that only summary data are available, or if a conservative correction is desired, LD Score regression intercept provides a more robust quantification of the extent of inflation from confounding bias than λ_{GC} . Since λ_{GC} increases with sample size in the presence of polygenicity (even without confounding bias), the gain in power obtained by correcting test statistics with the LD Score regression intercept instead of λ_{GC} will become even more substantial for larger GWAS. Therefore, LD Score regression provides a method less likely to be biased from confounding bias and polygenicity as compared to conventional observational studies (*i.e.*, phenotypic analysis, details for this method can be found in the original paper: Bulik-Sullivan B, Finucane HK et al., An atlas of genetic correlations across human diseases and traits, Nat Genet, 2015, PMID: 26414676). We have revised it accordingly in the Abstract. *“All analyses were less likely to be affected by reverse causation and selection bias as well as environmental and behavioral confounding as compared to conventional observational studies.”*

Your response seems to be about confounding bias, but I was talking very generally about selection bias into cohort studies. People in cohort studies tend to be healthier, wealthier, and younger (in general) – this can lead to collider bias in genetic association studies. I don't think you need to discuss this in any length, I'd just be tempted to remove the claim about selection bias.

5. – You talk about lung function decline throughout, but haven't you only looked at GWAS of cross-sectionally measured lung function? If you have used cross-sectional GWAS only, this is not studying lung function decline.

>>Response: Thank you for your suggestions, we have now revised as “lung function” throughout the manuscript.

Thank you!

6. – See also above. I am aware that you now cite this paper (<https://link.springer.com/article/10.1007/s10654-021-00726-8>), I did a ctrl+F for ‘Steiger’ in your supplement (427952_1_supp_8388198_s54xtc.pdf) and cannot find any added result as you mention in your response letter. Some tables are truncated at the right edge for me though, as the pages are all in portrait view – maybe it is a format issue. Also, what is the PMR-STEIGER that you refer to in the Results section of the main manuscript? Is this testing that the variance explained by the SNPs in the outcome is less than that explained in the exposure? If so, I would be more explicit about this. I would

be tempted to apply Steiger ‘filtering’ (i.e. dropping IVs that explain more variance in GlycA than LF) to the reverse direction MR. If you see no evidence of association after doing so, this would help you be more confident in the forward GlycA↔ LF association. As your results stand currently, you do observe strong effects in the reverse direction, so if the reverse direction MR withstands Steiger filtering, I would stress the possibility of bidirectionality more strongly.

>>Response: Thank you for your comments. We are sorry for the format issue in the supplementary files, we have gone through the supplementary files to make sure all the tables are in correct format.

We conducted Steiger directionality test in the previous revision and included p-values for Steiger directionality test in Supplementary table 24 (all p values were small suggesting that the overall direction of the observed MR associations was correct). Then, we conducted Steiger filtering to check if any of the instrument variables explained substantially more variance in the exposure than in the outcome, we found all SNPs in the forward direction survived Steiger filtering (we now added the related results in the Results Section and Supplementary files). In the reverse direction, we filtered out 11 SNPs (rs149717632, rs17427599, rs2858863, rs3135387, rs330925, rs342466, rs550057, rs56083805, rs62053264, rs62091797, rs695238), 13 SNPs (rs10737680, rs12132138, rs1299113, rs2927325, rs3130283, rs56349217, rs62091797, rs7092539, rs72738786, rs7613360, rs7947322, rs8126001, rs9411378), 9 SNPs (rs113638840, rs11759008, rs2414870, rs4987082, rs60648773, rs73390208, rs7896518, rs9264803, rs983309) and 7 SNPs (rs2260000, rs2352974, rs2571392, rs2844498, rs529565, rs56121115, rs713916, we only included p values for overall Steiger directionality test in the table to minimize the space for supplementary files) for FEV1, FVC, FEV1/FVC and PEF, respectively and repeated the reverse MR, our results showed consistent results with the primary GSMR and MR-PRESSO results before filtering (see table below for the results after Steiger filtering). This is because our primary MR analysis using GSMR have filtered pleiotropic SNPs using HEIDI. HEIDI is a stringent method to detect pleiotropic SNPs at which the estimates of b_{xy} (the estimate of the effect of x on y) are significantly different from expected under a causal model, and remove them from the GSMR analysis, with accounting for LD among SNPs using the reference dataset for LD estimation. We mainly observed consistent significant instrumental effect of FEV1 and FVC with glyca in the reverse direction. As suggested, we have revised the sentences accordingly in the manuscript. *“ Additionally, sensitivity analysis using MR_PRESSO showed consistent results with GSMR, and Steiger directionality test showed that all the causal estimates were oriented in the intended direction (i.e., all $P_{MR-Steiger} < 0.05$ suggesting that the overall direction of the observed MR associations were correct, Supplementary Table 24). Furthermore, MR results after Steiger filtering showed consistent results with GSMR and suggested the association of glyca with FEV1 and FVC are more likely to be bidirectional (Supplementary Table 25).”* (lines 395 to 401, page 19)

Thanks - I think at the moment there isn't anything very definitive on bidirectionality in the Discussion. A couple of sentences would be useful (but see earlier comment about editorial discretion).

Exposure	Outcome	Causal Estimate	Sd	T-stat	P-value
FEV1	glycA	-0.07	0.01	-4.90	1.31E-06
FVC	glycA	-0.10	0.02	-5.78	1.52E-08
FEV1_FVC_Ratio	glycA	0.05	0.01	4.66	3.98E-06
PEF	glycA	-0.01	0.01	-0.31	0.76
FEV1	CRP	0.00	0.00	-0.43	0.67
FVC	CRP	-0.02	0.01	-3.04	2.57E-03
FEV1_FVC_Ratio	CRP	0.00	0.00	0.34	0.73
PEF	CRP	0.00	0.01	-0.21	0.83
FEV1	albumin	0.00	0.01	-0.05	0.96
FVC	albumin	0.01	0.01	0.50	0.62
FEV1_FVC_Ratio	albumin	-0.01	0.01	-1.07	0.28
PEF	albumin	-0.01	0.01	-1.33	0.19

7. - Re: [[Response: Thank you for your great suggestions. According to your suggestion, we have added one sentence in the Discussion section to make clear that some of our corrections may be overly stringent due to high correlation between traits we included. “Fourthly, some of our analysis may suffer from multiple testing burden, because traits included in the present study, such as lung function parameters were highly correlated.”]]

- I would edit to say ‘some of our multiple testing corrections may be overly stringent’

>>Response: Thank you for your suggestion. We have revised it as “Fourthly, some of our multiple testing corrections may be overly stringent...” .

Great 😊

REVIEWER COMMENTS

Reviewer #2 (Remarks to the Author):

1. Thank you for your response. My concern was not whether GlycA is a true biomarker for inflammation, but more how specific it would be in predicting respiratory disease versus many other disease (even ignoring any association in the reverse direction).

>>Response: Thank you so much for your comments. We agree the specificity of GlycA as either a drug target or biomarker is of great importance and we have added this as a limitation in the Discussion section.

“Thirdly, although we found potential causality and shared mechanisms of GlycA on lung function and onset of asthma, the clinical utility, specificity and potential therapeutic value and of GlycA on lung function and respiratory diseases are warranted for further validation as it is a composite measure reflecting changes in the number and the complexity of N-glycan side chains of acute phase reactant proteins.”

2. Thank you - so as you state, you do find evidence of bidirectionality in your analyses (e.g. GlycA may cause lung function, but equally, lung function may change GlycA). I would have preferred that you had made more of this in the Discussion, but I will leave it to editorial discretion as to whether that is appropriate.

>>Response: Thank you so much for your comments. we have added more discussion on the bidirectionality of GlycA with lung function parameters and respiratory diseases.

“ Additionally, reverse MR showed significant negative effects of FEV1 and FVC on GlycA suggesting that improvement in lung function could also be a potential indicator of systematic inflammation.”

3. The value of -0.84 is taken from a genetic correlation of the Shrine et al 2019 FEV1/FVC data, and the Sakornsakolpat et al 2019 COPD data. But it is nevertheless reassuring that you see a correlation between COPD and FEV1/FVC.

>>Response: Thank you so much.

4. I agree - the correlation you observe is entirely consistent with that observed in Shrine et al 2019 - it is helpful to know that.

>>Response: Thank you so much.

5. Thank you for all of this extra work. I remain struck by the fact that you see a more significant genetic correlation between GlycA and COPD than you do with GlycA and FEV1/FVC, because I would expect the power to be greater for the FEV1/FVC GWAS (as this is a quantitative trait), unless the sample size for the COPD GWAS is enormous I wonder if it's possible that the COPD GWAS is capturing something else, e.g. smoking, which would be associated with inflammation, and that this is driving the correlation

(you may have thoughts on this)? Anyway, I think all you can do is (as you have done), be frank about what you have found.

>>Response: Thank you so much for your comments. Yes, it is possible that the COPD GWAS may capture something else like smoking, diet which related with inflammation.

6. Your response seems to be about confounding bias, but I was talking very generally about selection bias into cohort studies. People in cohort studies tend to be healthier, wealthier, and younger (in general) - this can lead to collider bias in genetic association studies. I don't think you need to discuss this in any length, I'd just be tempted to remove the claim about selection bias.

>>Response: Thank you for your suggestions. We have removed this sentence from abstract as suggested.

7. Thanks - I think at the moment there isn't anything very definitive on bidirectionality in the Discussion. A couple of sentences would be useful (but see earlier comment about editorial discretion).

>>Response: Thank you for your suggestions. We have added two sentences in the manuscript to make it clear that we observed bi-directional genetic effects of GlycA with lung function.

"Bi-directional GSMR suggested that GlycA was causally related with lung function and vice versa." and *"Additionally, reverse MR showed significant negative effects of FEV1 and FVC on GlycA suggesting that improvement in lung function could also be a potential indicator of systematic inflammation."*